# A synergistic exploitation to produce high-voltage quasi-solid-state lithium metal batteries

Junru Wu[1,2], Xianshu Wang[1,2], Qi Liu[1,2], Shuwei Wang[1,2], Dong Zhou [1,2,3✉], Feiyu Kang[1,2], Devaraj Shanmukaraj [4], Michel Armand [4✉], Teofilo Rojo[5], Baohua Li [1,2✉] & Guoxiu Wang [3✉]

The current Li-based battery technology is limited in terms of energy contents. Therefore, several approaches are considered to improve the energy density of these energy storage devices. Here, we report the combination of a heteroatom-based gel polymer electrolyte with a hybrid cathode comprising of a Li-rich oxide active material and graphite conductive agent to produce a high-energy "shuttle-relay" Li metal battery, where additional capacity is generated from the electrolyte's anion shuttling at high voltages. The gel polymer electrolyte, prepared via in situ polymerization in an all-fluorinated electrolyte, shows adequate ionic conductivity (around 2 mS cm$^{-1}$ at 25 °C), oxidation stability (up to 5.5 V vs Li/Li$^+$), compatibility with Li metal and safety aspects (i.e., non-flammability). The polymeric electrolyte allows for a reversible insertion of hexafluorophosphate anions into the conductive graphite (i.e., dual-ion mechanism) after the removal of Li ions from Li-rich oxide (i.e., rocking-chair mechanism).

[1] Tsinghua Shenzhen International Graduate School, Tsinghua University, Shenzhen 518055, China. [2] School of Materials Science and Engineering, Tsinghua University, Beijing 100084, China. [3] Centre for Clean Energy Technology, School of Mathematical and Physical Sciences, Faculty of Science, University of Technology Sydney, Sydney, NSW 2007, Australia. [4] Centre for Cooperative Research on Alternative Energies (CIC energiGUNE), Basque Research and Technology Alliance (BRTA), Alava Technology Park, Albert Einstein 48, 01510 Vitoria-Gasteiz, Spain. [5] Inorganic Chemistry Department, University of the Basque Country UPV/EHU, Bilbao 48080, Spain. ✉email: zhoudong087@gmail.com; marmand@cicenergigune.com; libh@sz.tsinghua.edu.cn; Guoxiu.Wang@uts.edu.au

Lithium (Li)-based batteries, particularly Li-ion batteries, have dominated the market of portable energy storage devices for decades[1]. However, the specific energy of Li-ion batteries is approaching their theoretical limit (300 Wh kg$^{-1}$), making it difficult to satisfy the requirement for long-distance driving with a single charging of electric vehicles[2].

To further increase the energy density of Li-based batteries, the upgrading of electrode and electrolyte materials is urgently desired. As for anode materials, Li metal has been regarded as the ideal candidate due to its specific capacity (3860 mAh g$^{-1}$) and the lowest redox potential ($-3.04$ V vs. standard hydrogen electrode)[3]. However, its practical application has been severely hampered by uncontrollable Li dendrite growth during cycling[4]. It is well-recognized that the highly reactive Li metal is prone to react with the electrolytes and form a passivated solid electrolyte interphase (SEI) layer on the surface[5]. Nevertheless, the strength of such an SEI layer generally cannot withstand the repeated volume changes during Li deposition and stripping, which results in surface defects and subsequent dendrite growth from these defects[6]. The resulting Li dendrites cannot only pierce through the separator and trigger catastrophic safety hazards but also constantly consume both active Li and electrolyte, giving rise to low Coulombic efficiency and degraded cycle life[7].

On the cathode side, layered transition metal oxides, e.g., nickel-rich oxides (LiNi$_{1-x}$M$_x$O$_2$, M = Co, Mn, and Al) and Li-rich oxides (LROs) (Li$_{1+x}$M$_{1-x}$O$_2$, M = Mn, Ni, and Co), are desirable for high-energy Li-based batteries considering their combined merits in specific capacity, working potential, and cycling performance[8]. However, for the intercalation-based Li-ion batteries, only the Li ions in the electrolyte participate in the electrochemical reactions based on a "rocking-chair" mechanism, while no extra capacity contribution is made by the anions in electrolytes. Therefore, unlocking the additional potential of anions in the electrolyte is a promising approach to further enhance battery energy density. Recently, dual-ion batteries (DIBs) based on graphitic cathode materials have attracted extensive attentions, in which anions (e.g., hexafluorophosphate (PF$_6^-$)[9], bis(trifluoromethanesulfonyl) imide (TFSI$^-$)[10] or bis(-fluorosulfonyl)imide (FSI$^-$)[11]) reversibly intercalate into/deintercalated from graphite interlayers at cell voltage >4.5 V during charge/discharge processes[12]. The operating voltage of these DIBs generally is about 5 V vs. Li/Li$^+$, which is favorable for energy density improvement[13]. However, such a high intercalation voltage of graphite leads to severe oxidative decomposition of the electrolytes and tends to construct a high-resistance cathode electrolyte interphase (CEI) on the cathode surface[14]. This seriously impedes anion insertion, resulting in inferior reversibility and poor cycling stability[15]. Furthermore, the co-intercalation of the solvent molecules into graphite cathode causes exfoliation of graphite layers and the subsequent irreversible loss of active materials during cycling[16]. As for the electrolytes, the flammable solvents (e.g., organic carbonates and ethers) widely applied in Li-based batteries trigger safety concerns including fire, explosion, and leakage of toxic electrolyte components[17]. All these drawbacks have brought great challenges for the development of high-energy Li-based batteries.

Here, we demonstrate that a highly reversible insertion/extraction of PF$_6^-$ anions between graphite interlayers can be achieved in a heteroatom-based gel polymer electrolyte (HGPE), which was synthesized via in situ copolymerization of diethyl allyl phosphate (DAP) monomer and pentaerythritol tetraacrylate (PETEA) crosslinker in the presence of an all-fluorinated electrolyte. This HGPE exhibited high safety (i.e., nonflammability and non-leakage), high ionic conductivity (1.99 mS cm$^{-1}$ at 25 °C), wide electrochemical window (up to 5.5 V vs. Li/Li$^+$), and compatibility with both Li metal anode (a Li deposition/stripping Coulombic efficiency of 99.7%) and graphite cathode (93% capacity retention after 1000 cycles). On this basis, we developed a "shuttle-relay" Li metal battery (SRLMB) consisting of a hybrid cathode with LRO as active material and KS6 graphite as a conductive agent and the HGPE as electrolyte. During the charge process, a reversible insertion of PF$_6^-$ anions into the KS6 graphite occurs after the stripping of Li ions from the LRO, in which anions contributes 8.2% (i.e., 3.2 Wh L$^{-1}$) extra energy density of the cell. The as-developed SRLMB exhibited high capacity and cycling stability, ascribed to the stable electrode|HGPE interfaces.

## Results and discussion

**Mechanism of "shuttle-relay" Li metal battery.** Figure 1a illustrates the working mechanism of the quasi-solid-state SRLMB, which is realized by the well-designed HGPE. Currently, Li-ion batteries extensively apply organic electrolytes containing cyclic carbonate solvents (e.g., ethylene carbonate (EC)) with high dielectric constant to dissolve lithium hexafluorophosphate (LiPF$_6$) salt, and linear carbonate solvents (e.g., ethyl methyl carbonate (EMC)) to reduce the electrolyte viscosity[11]. However, such carbonate-based electrolytes generally show poor compatibility with both Li metal anode and 5 V-class cathodes (e.g., LRO for "rocking-chair" chemistry and graphitic carbon for "dual-ion" chemistry). On the anode side, the electrolyte solvents cannot form a stable SEI layer on the Li metal surface, leading to Li dendrite growth and low Columbic efficiency[18]. On the cathode side, the insufficient oxidation resistance of carbonate solvents triggers severe electrolyte decomposition and constructs a thick CEI with high resistance, which dramatically degrades the battery performance (Fig. 1a, upper panels). The interfacial issues are even more severe for graphite cathodes, since the carbonate molecules tend to co-intercalate into graphite interlayers, giving rise to low reversible anions insertion/extraction capacity accompanying structural deterioration[16].

To address these issues, we developed an all-fluorinated electrolyte for high-voltage Li metal batteries, which contains 1 M LiPF$_6$ dissolved in a mixture of fluoroethylene carbonate (FEC): 2,2,2-trifluoroethylmethyl carbonate (FEMC): 1,1,2,3,3,3-hexafluoropropyl-2,2,2-trifluoroethylether (HTE) with a volume ratio of 1: 6: 3. The effect of salt concentration on electrolyte ionic conductivity and battery performance were discussed in Supplementary Fig. 1 and Supplementary Note 1. In this liquid electrolyte, FEC is beneficial for improving the compatibility with Li anodes, meanwhile, FEMC ensures a reversible insertion/extraction of PF$_6^-$ into graphite (Supplementary Fig. 2). Moreover, as a novel electrolyte component, the HTE not only functions as a diluent to reduce the electrolyte viscosity, but also optimizes the localized solvation structure of cation/anion aggregates, thus further stabilizing the Li|electrolyte interfaces. On this basis, 3 wt% DAP monomer and 1.5 wt% PETEA crosslinker were in situ polymerized in this fluorinated electrolyte to form an HGPE, in which the three-dimensional polymer matrix effectively improves the electrolyte safety by preventing liquid leakage (Supplementary Fig. 3). To verify the effect of each electrolyte component, we carried out density functional theory (DFT) calculation on the highest occupied molecular orbital (HOMO) and lowest unoccupied molecular orbital (LUMO) energies of the solvent molecules. Based on molecular orbital theory, the HOMO energy correlates to the oxidative decomposition potential, while the LUMO energy is associated with the reductive decomposition potential[19]. As shown in Fig. 1b, the HOMO energy of fluorinated solvents (i.e., FEC: $-7.3278$ eV, FEMC: $-7.001$ eV, and HTE: $-8.1353$ eV) is much lower than those of EC ($-6.8853$ eV) and EMC ($-6.4791$ eV), demonstrating the superior oxidation resistance of fluorine solvents owing to the

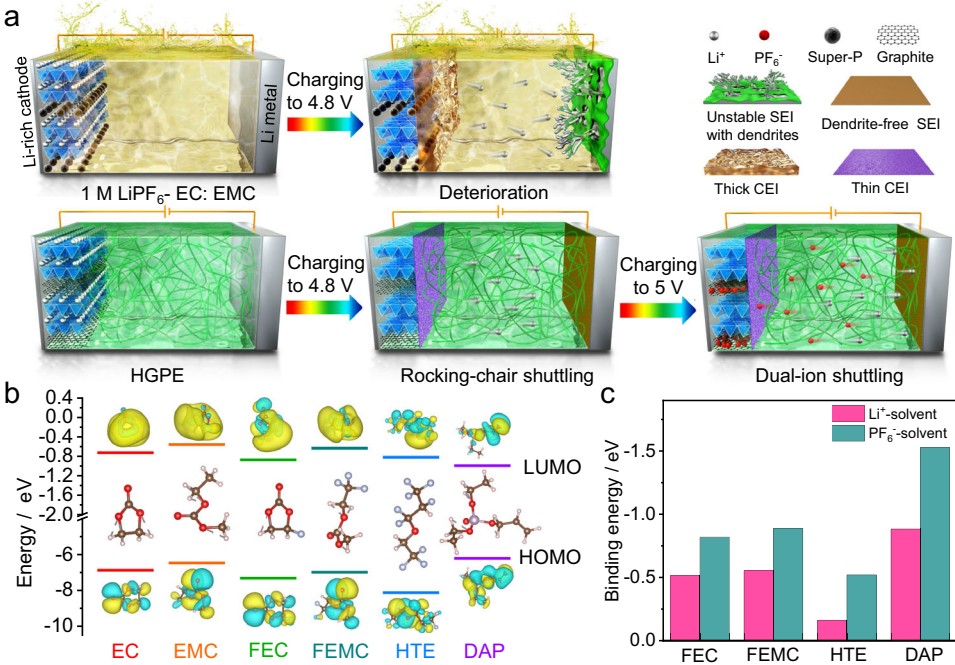

**Fig. 1 The design of HGPE. a** Schematic illustration of the mechanisms of a Li|1 M LiPF$_6$-EC: EMC | LRO "rocking-chair" battery (upper panels) and a "shuttle-relay" battery with a hybrid LRO cathode using graphite as a conductive agent, a Li metal anode, and an HGPE (lower panels). **b** The LUMO and HOMO energy values of the solvent molecules. The molecular structures and corresponding visual LUMO and HOMO geometry structures are shown as insets. Brown, white, red, purple, and blue balls represent carbon, hydrogen, oxygen, phosphorus, and fluorine atoms, respectively. **c** Binding energies of FEC, FEMC, HTE, and DAP for a Li$^+$ cation and a PF$_6^-$ anion.

strong electron-withdrawing effect of fluorine atoms on the core of solvent molecules[20]. Meanwhile, the DAP monomer presents the highest HOMO value (−6.2292 eV). As a result, the residual DAP monomer after polymerization acts as a CEI-forming additive in the HGPE to further inhibit the electrolyte oxidation and the co-intercalation of solvent molecules into graphite (Supplementary Fig. 4). Moreover, FEC and DAP exhibit the lowest LUMO energies of −0.8835 and −1.0021 eV, respectively. Consequently, FEC and the residual DAP monomer will be preferentially reduced on the Li anode to form a protective LiF-rich and phosphorus-containing SEI inhibiting dendrites growth. To evaluate the effect of HTE, we calculated the binding energy of electrolyte components with Li$^+$ cation and PF$_6^-$ anion (Fig. 1c). It is seen that the HTE shows the lowest absolute values of binding energy with both Li$^+$ and PF$_6^-$, indicating a weak interaction between HTE and ions. This is consistent with the lower solubility of LiPF$_6$ salt in HTE compared with other solvents (Supplementary Fig. 5). Therefore, the introduction of HTE enables the formation of a highly concentrated electrolyte in local regions by increasing the ratio of ion: fluorinated carbonate in the solvation structures. Such a unique solvation structure can minimize the excessive side reactions between electrolyte solvents and electrodes, thus improving the battery performance[21].

When applied in SRLMBs, the HGPE enables a characteristic mechanism (i.e., the shuttle-relay), which synergistically exploits the LRO's rocking-chair and the graphite's dual-ion mechanisms. As shown in the lower panels of Fig. 1a, Li ions are stripped from the LRO cathode in the voltage range of 2.0–4.8 V, followed by insertion of the PF$_6$ anions into the conductive graphite at 4.8–5.0 V. In the HGPE, the synergistic effect of liquid-state electrolyte components constructs robust SEI/CEI to improve the electrode|electrolyte compatibility; meanwhile, the polymer matrix efficiently retards the migration of Li$^+$/PF$_6^-$ ions to SEI/CEI surface defects through strong interaction (see the high binding energy absolute values between DAP and Li$^+$/PF$_6^-$,

Fig. 1c), thereby favoring a uniform Li$^+$/PF$_6^-$ ion flux to promote uniform Li insertion and anions intercalation into graphite[7].

**Characterization of the HGPE.** As seen from Fig. 2a, the as-synthesized HGPE appeared as a non-flowing white gel. According to $^1$H NMR, the conversion of PETEA and DAP monomer in HGPE were 81.9 and 32.4%, respectively (Supplementary Fig. 6 and Supplementary Note 2)[22]. Figure 2b exhibits the Fourier transform infrared spectra (FTIR) of DAP monomer, PETEA crosslinker, and HGPE polymer matrix. The characteristic peaks at around 1013 cm$^{-1}$ (P-O stretching), 1099 cm$^{-1}$ (P=O stretching), 1260 cm$^{-1}$ (C-O antisymmetric stretching), 1470 cm$^{-1}$ and 1406 cm$^{-1}$ (CH$_2$ bending), and 1720 cm$^{-1}$ (C=O stretching) presented in the FTIR spectra of DAP and PETEA[23,24]. Upon polymerization, the absorption peak at about 1630 cm$^{-1}$ correlated to the stretching vibration of the C=C bond nearly disappeared in the polymer matrix, indicating a high-degree polymerization of monomer and crosslinker in the HGPE. Figure 2c demonstrates the combustion tests of 1 M LiPF$_6$-EC: EMC electrolyte and the HGPE. This result is well-consistent with the FTIR curve of precursor solution and HGPE (Supplementary Fig. 7). It is seen that the traditional liquid electrolyte easily caught fire on the ignition, and kept on burning even after removing the torch with a self-extinguishing time (SET) of 92 s g$^{-1}$ (Supplementary Movie 1). This significantly differs from the HGPE which exhibited zero SET after removal of the torch, indicating its excellent nonflammability (Supplementary Movie 2). This is attributed to facts that the substitution of hydrogen atoms by fluorine atoms in fluorinated solvents significantly reduces the generation of hydrogen radicals, which hence diminishes combustion hazard (Supplementary Fig. 8 and Supplementary Movies 3, 4), and the thermal decomposition products of DAP at a high temperature can further capture the radicals and prevent the unwanted combustion chain reactions[25]. Moreover, it is worth noticing that the polymer matrix in the

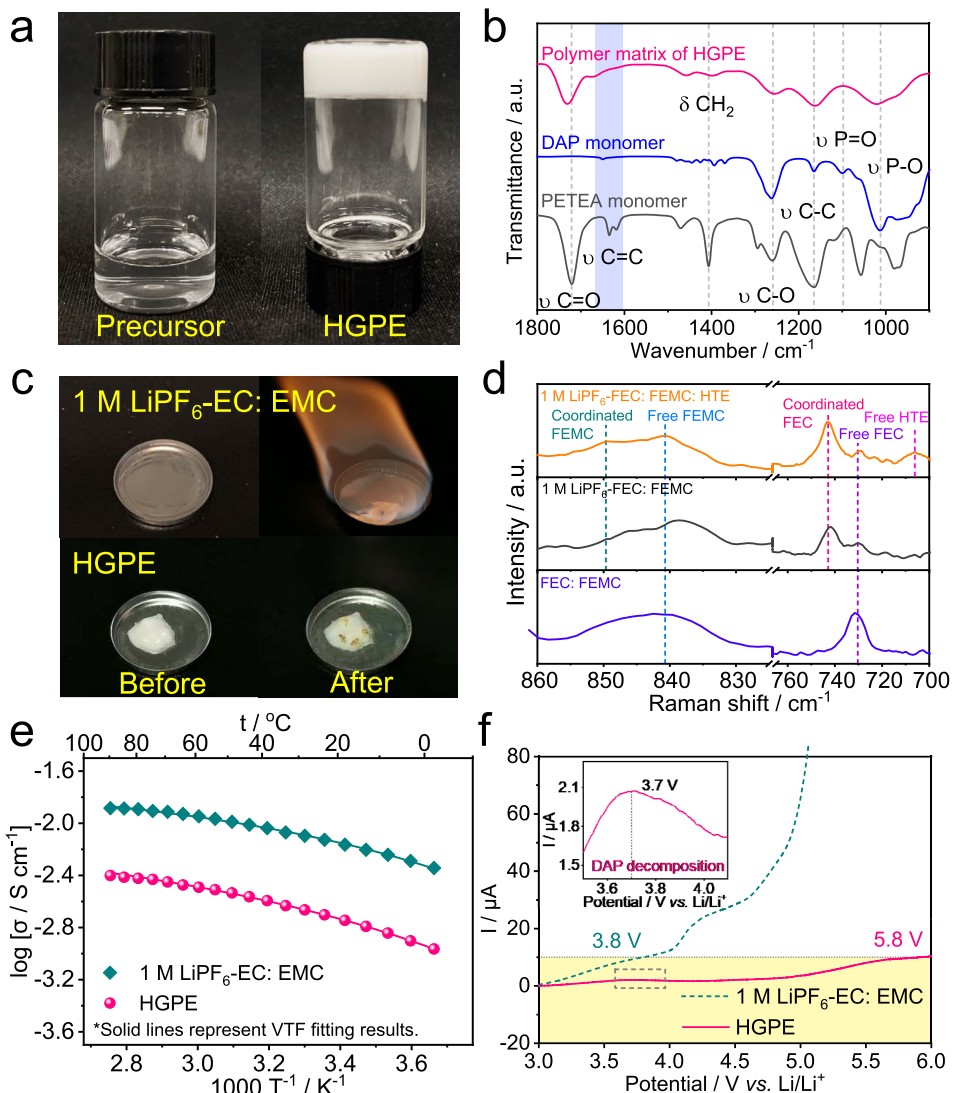

**Fig. 2 Characterization of the HGPE. a** Optical images of the precursor solution (left) and the corresponding HGPE (right) after copolymerization; **b** FTIR spectra of the DAP, PETEA, and the polymer matrix of HGPE; **c** Combustion tests of 1 M LiPF₆-EC: EMC electrolyte and HGPE; **d** Raman spectra of FEC: FEMC mixture, 1 M LiPF₆-FEC: FEMC and 1 M LiPF₆-FEC: FEMC: HTE electrolytes; **e** Ionic conductivities for 1 M LiPF₆-EC: EMC electrolyte and HGPE at various temperatures from 0–90 °C. The plots represent the experimental data while the solid lines represent VTF fitting results. **f** LSV curves of 1 M LiPF₆-EC: EMC electrolyte and HGPE at a scan rate of 5 mV s⁻¹, using platinum foil as the working electrode and Li foil as the counter and reference electrodes.

HGPE effectively immobilizes the solvents and decreases their volatility, thus preventing the risk of liquid leakage (Supplementary Figs. 9, 10)[9,26]. Such high safety of the HGPE is a critical asset for the practical application of high-energy Li metal batteries.

Raman spectra was measured to characterize the coordination environment in the electrolyte. It is seen that in the mixture of FEC: FEMC (1: 6 by volume), peaks at around 730 cm⁻¹ were recorded (assigned to free FEC) and one at about 840 cm⁻¹ corresponding to the free FEMC molecule (Fig. 2d and Supplementary Fig. 11). After dissolving 1 M LiPF₆ into the mixture, the peak intensity of free solvent molecules diminished accompanying the appearance of new bands at about 849 cm⁻¹ (Li⁺-coordinated FEMC), 921 and 745 cm⁻¹ (Li⁺-coordinated FEC)[27]. With the addition of HTE, an extra peak of free HTE molecules was observed at about 706 cm⁻¹ in the spectrum of LiPF₆-FEC: FEMC: HTE electrolyte, meanwhile the peak intensity of Li⁺-coordinated carbonates increased, which verifies that more fluorinated carbonate molecules are coordinated with Li ions in

the solvation sheaths[28]. This is well-consistent with the binding energy calculation results in Fig. 1c, which efficiently alleviates the excessive side reactions between free solvent molecules and Li metal.

Ionic conductivity is considered as an important property of electrolytes. Figure 2e and Supplementary Fig. 12 show the temperature dependences of ionic conductivities for 1 M LiPF₆-EC: EMC (1:6 by volume), 1 M LiPF₆-FEC: FEMC (1:6 by volume), 1 M LiPF₆-FEC: FEMC: HTE (1:6:3 by volume), and HGPE electrolytes within the temperature range 0 to 90 °C. The plot of log σ vs. T⁻¹ for electrolytes presents a nonlinear relationship that can be well fitted by the following empirical Vogel–Tamman–Fulcher (VTF) formula[29]:

$$\sigma = \sigma_o T^{\frac{-1}{2}} \exp\left(-\frac{E_a}{R(T - T_o)}\right) \quad (1)$$

where $\sigma_o$ is the preexponential coefficient, $E_a$ is the pseudo-activation energy, $T_o$ is the parameter related to the ideal glass transition temperature, and $R$ is the gas constant. The fitted

parameters and ionic conductivity values are presented in Supplementary Table 1. It is seen that the 1 M LiPF$_6$-FEC: FEMC and 1 M LiPF$_6$-FEC: FEMC: HTE electrolytes showed a slight decrease in ionic conductivity at 25 °C due to the high viscosity of the fluorinated solvents. After the in situ gelation, the ionic conductivity of HGPE maintained 1.99 mS cm$^{-1}$ at 25 °C. Meanwhile, the $E_a$ value ($1.00 \times 10^{-2}$ eV) was very close to that of the 1 M LiPF$_6$ in FEC: FEMC: HTE liquid electrolyte ($9.30 \times 10^{-3}$ eV). This indicates that the hindrance of the gel matrix for wanted ion transport is negligible. Such a high ionic conductivity value of the HGPE enables efficient battery operation at high C rates.

The electrochemical stability window of the electrolytes was investigated by linear sweeping voltammetry (LSV). As shown in Fig. 2f, a low oxidation current was observed until 5.8 V for HGPE. The high electrochemical stability of HGPE mainly originates from the fluorination of the electrolyte solvent and the robust CEI formed by the oxidation of DAP at about 3.7 V (inset of Fig. 2f), which enables its application in 5 V-class Li metal batteries. In sharp contrast, the irreversible oxidation voltages of 1 M LiPF$_6$-EC: EMC, 1 M LiPF$_6$-FEC: FEMC, and 1 M LiPF$_6$-FEC: FEMC: HTE were around 3.8, 4.6, and 5.2 V vs. Li/Li$^+$ (Fig. 2f and Supplementary Fig. 13), respectively. Furthermore, the PETEA-DAP polymer framework can restrict the movement of anions, resulting in an increased Li-ion transfer number ($t_{Li}^+$) for the HGPE (i.e., 0.43) compared with the 1 M LiPF$_6$-FEC: FEMC: HTE electrolyte (0.37, Supplementary Fig. 14 and Supplementary Table 2)[9]. Such increased $t_{Li}^+$ is close to 0.5, which facilitates the balance of the active ions in the DIBs[30].

**Li metal morphology and interfacial chemistry**. To investigate the stability of Li metal anodes in different electrolytes, the voltage variation of symmetric Li||Li cells was assessed during galvanostatic cycling at a constant current of 0.5 mA cm$^{-2}$. As shown in the inset of Fig. 3a, the Li|Li symmetric cell using 1 M LiPF$_6$-EC: EMC electrolyte exhibited a dramatically increased overpotential with cycling time (around 5 V at 420 h), mainly due to the thickening of the SEI layer and continuous Li dendrite growth[18]. The overpotential of Li|1 M LiPF$_6$-FEC: FEMC|Li cells and Li|1 M LiPFF$_6$-FEC: FEMC: HTE|Li cells were about 120 and 150 mV, and the cells failed at 600 and 800 h, respectively (Supplementary Fig. 15). Meanwhile, the Li|HGPE|Li cell delivered a stable voltage hysteresis of 100 mV with no oscillation throughout a 1400 h cycling (the general decrease in overpotential in the initial cycles is related to the activation of Li anode with a pristine oxide layer on the surface[31]), indicating a dendrite-free Li deposition when using HGPE. The temperature-dependent electrochemical impedance spectroscopy (EIS) measurements of the Li||Li cells after 20 cycles were carried out to further evaluate the activation energies during Li deposition/stripping. The activation energies derived from the SEI ($R_{sei}$) and ion transfer resistance ($R_{ct}$) are denoted as $E_{a1}$ and $E_{a2}$, corresponding to the energy barriers for Li ions transport across the SEI layer and their desolvation from Li$^+$ solvation shells, respectively (Supplementary Tables 3–6)[32]. The $E_{a1}$ for HGPE (46.07 kJ mol$^{-1}$) is significantly lower than those for the 1 M LiPF$_6$-EC: EMC (76.65 kJ mol$^{-1}$), 1 M LiPF$_6$-FEC: FEMC (50.63 kJ mol$^{-1}$), and 1 M LiPF$_6$-FEC: FEMC: HTE (59.86 kJ mol$^{-1}$) electrolytes (Fig. 3b and Supplementary Fig. 16). This implies that the structure and composition of the SEI layer formed in the presence of HGPE endows a fast Li ion transport kinetics. Moreover, although the $E_{a2}$ value for HGPE (58.28 kJ mol$^{-1}$) is slightly higher than that for 1 M LiPF$_6$-EC: EMC electrolyte (51.13 kJ mol$^{-1}$) due to the stronger interaction between PF$_6^-$-EC than that between PF$_6^-$-fluorinated carbonate that facilitates Li$^+$ desolvation from the ion pairs and

aggregates[33,34], the $E_{a2}$ for HGPE is still lower than that for 1 M LiPF$_6$-FEC: FEMC electrolyte (62.52 kJ mol$^{-1}$) and 1 M LiPF$_6$-FEC: FEMC: HTE electrolyte (69.67 kJ mol$^{-1}$, Supplementary Figs. 16, 17). This could be attributed to the fact that a DAP-based polymer matrix can promote the dissociation of Li ions from the solvation sheath[35]. The low $E_{a1}$ and $E_{a2}$ of the HGPE-based cell promote a low-resistance Li|HGPE interface with fast ion diffusion and conversion.

The average Li plating/stripping Coulombic efficiency (CE$_{avg}$) measurement was further conducted in Li||Cu cells in various electrolytes[36]. The cells using the HGPE exhibited a CE$_{avg}$ of 99.7%, which could be considered as an appealing experimental result compared with state-of-the-art electrolytes[6,37,38] and 1 M LiPF$_6$-EC: EMC (69.9%), 1 M LiPF$_6$-FEC: FEMC (98.4%), and 1 M LiPF$_6$-FEC: FEMC: HTE electrolyte (98.6%, Fig. 3c and Supplementary Fig. 18). The plating morphologies of Li on Cu substrates were examined by field emission scanning electron microscope (FE-SEM). As shown in Fig. 3d, the Li|1 M LiPF$_6$-EC: EMC|Cu cell present a highly loose and mossy deposition structure with a thickness of 29.0 μm, far exceeding the theoretical value (about 9.7 μm). After fluorinating the carbonate solvents and introducing the HTE diluent, the surfaces of deposited Li gradually became smoother and denser, and the thicknesses of Li deposition decreased to around 21.4 and 16.1 μm, respectively (Supplementary Fig. 19). In the Li|Cu cell employing the HGPE, for comparison, the plating Li showed a compact morphology as aggregated large particles, and the plated thickness (around 10.7 μm) was very close to the theoretical value (Fig. 3e). Such a dense Li deposition with a smaller surface/volume ratio effectively minimizes the parasitic reaction between metallic Li and electrolyte, and thus enables the high CE$_{avg}$ of Li|HGPE|Cu cells.

To analyze the microstructure of the SEI, 1 mAh cm$^{-2}$ Li was repeatedly plated on and stripped off a Cu grid for ten cycles at 0.2 mA cm$^{-2}$ to obtain a vacant SEI shell for morphology characterization. The transmission electron microscopy (TEM) images are shown in Fig. 4a, b. It is seen that a large amount of "dead Li" (i.e., electronically disconnected) residues appeared on the Cu grid cycled in 1 M LiPF$_6$-EC: EMC electrolyte, indicating irreversibility of Li plating/stripping (Fig. 4a, inset). The SEI was mainly composed of Li$_2$O particles distributed in an amorphous matrix (Fig. 4a), and the composition was further identified as organic compounds (e.g., ROCO$_2$Li, where "R" represents functional groups) originating from the decompositions of carbonate solvents[32], and Li$_x$PO$_y$F$_z$/Li$_2$O as the decomposition products of LiPF$_6$ salt, respectively (see the in-depth X-ray photoelectron spectroscopy (XPS) results in Supplementary Figs. 20–23)[39]. The Young's modulus of this SEI layer was as low as 398 MPa (Fig. 4c). For comparison, the amount of residual inactive Li obviously decreased on the Cu substrates in fluorinated electrolytes (Supplementary Fig. 24). Meanwhile, the proportion of LiF, mainly originating from the reduction of fluorinated solvents, greatly increased in the SEI (Supplementary Figs. 20–23). It is well-known that LiF with high mechanical strength (i.e., a shear modulus of 55.1 GPa, almost 11 times higher than that of Li metal (4.9 GPa)) can significantly enhance the robustness and interfacial energy of SEI layers, thus blocking Li dendrite growth[25]. As expected, the mechanical strength of SEIs in 1 M LiPF$_6$-FEC: FEMC and 1 M LiPF$_6$-FEC: FEMC: HTE electrolytes increased to 911 and 1426 MPa, respectively (Supplementary Figs. 25, 26). In the Cu grid retrieved from the cell employing the HGPE, a high Young's modulus up to 2768 MPa (Fig. 4d) has been achieved, owing to the coexistence of LiF and phosphorous-containing compounds (i.e., P-O-C and P=O) derived from the residual DAP monomer in the SEI (Fig. 4b and Supplementary Figs. 20–23). The robust SEI can efficiently suppress the formation of Li dendrite and dead Li (Fig. 4b, inset)

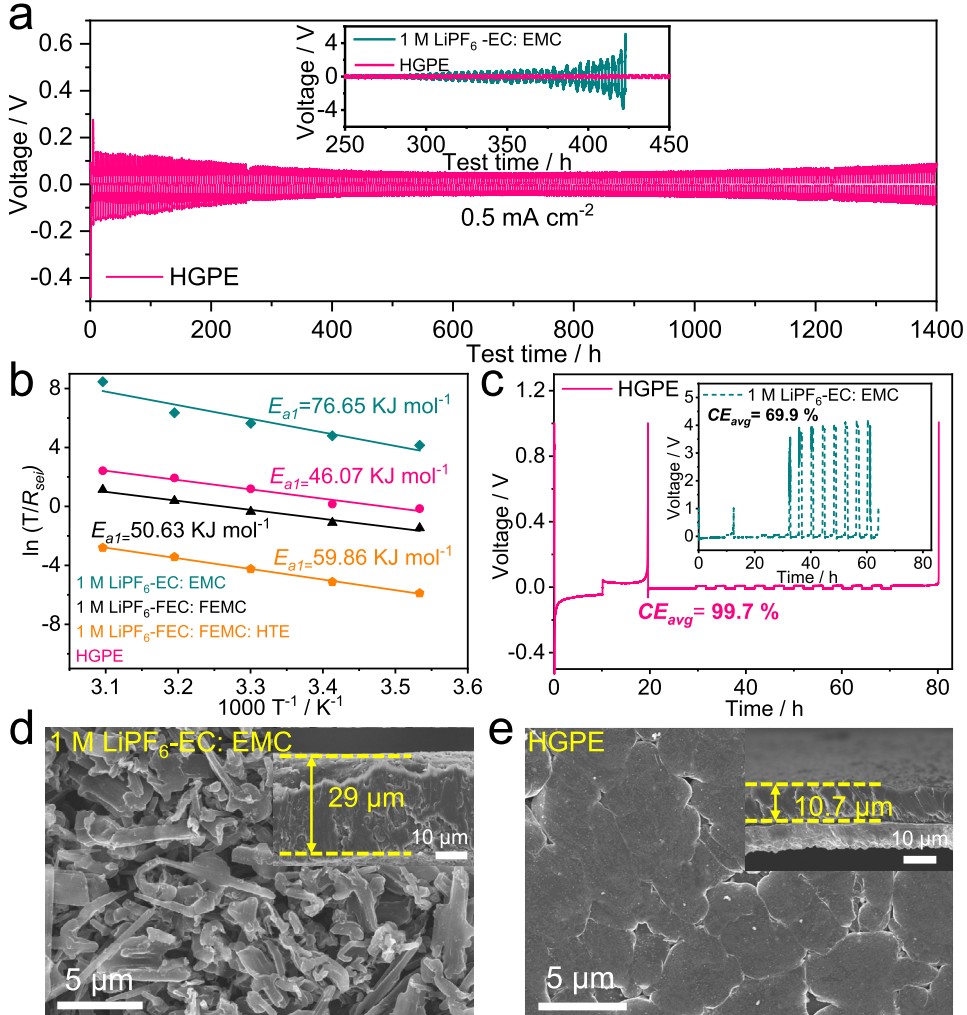

**Fig. 3 Lithium plating/stripping behavior in various electrolyte formulations. a** Voltage profiles of Li||Li symmetric cells using 1 M LiPF$_6$-EC: EMC electrolyte (shown in inset) and HGPE at 0.5 mA cm$^{-2}$ with a cutoff capacity of 1 mAh cm$^{-2}$; **b** The activation energies of $R_{sei}$ derived from Nyquist plots; **c** CE$_{avg}$ tests of Li plating-stripping in Li|1 M LiPF$_6$-EC: EMC | Cu (shown in inset) and Li|HGPE|Cu cells at 0.5 mA cm$^{-2}$ with a capacity of 1 mAh cm$^{-2}$; **d**, **e** Top and cross-sectional (shown in inset) FE-SEM images of the Li deposition obtained by plating 1 mAh cm$^{-2}$ Li on Cu substrate at 0.2 mA cm$^{-2}$ in Li||Cu cells using **d** 1 M LiPF$_6$-EC: EMC electrolyte and **e** HGPE. Scale bars: 5 μm in Fig. 3d, e; 10 μm in the inset of Fig. 3d, e.

and leads to a smooth surface morphology of the cycled Cu grid (Fig. 4d, inset). The above results are well-consistent with the electrochemical behavior in traditional Li metal batteries.

The stabilization effect of HGPE on the Li|electrolyte interface can be elucidated as follows. It is known that the ion transfer kinetics and Li deposition behavior are mainly determined by the composition and morphology of the SEI[32]. As shown in Fig. 4e, in the traditional 1 M LiPF$_6$-EC: EMC electrolyte, the Li$^+$ solvation shell consists of large amounts of carbonate solvent molecules but with negligible PF$_6^-$ anions solvation[40]. Upon electrochemical reaction, EC and EMC molecules in the solvation shell will be reduced and constitute the main component of SEI. Such an organic component (e.g., ROCO$_2$Li)-rich SEI is of insufficient strength to inhibit the dendrite growth, and repeatedly breaks down/reconstructs during the cycling, which causes raised thickness and resistance. This gives rise to a large energy barrier ($E_{a1}$) for Li ions to transport through the SEI, triggers inhomogeneous charge distribution, and aggravated polarization, further promoting dendrite formation[4]. In contrast, in HGPE, although the activation energy for dissociating the Li ions from the solvation sheath ($E_{a2}$) is similar to that for the traditional liquid electrolyte, the $E_{a1}$ is significantly reduced to facilitate Li

ion diffusion through the SEI. This is because the addition of HTE as diluent leads to a formation of localized highly concentrated regions in the electrolyte, in which fluorinated carbonates and Li$^+$-anion ion pairs participate in the solvation shell[32]. This solvation shell structure and residual DAP monomer in the HGPE endow a formation of an inorganic component (e.g., LiF-rich) SEI on the Li metal surface, which is highly robust to suppress dendrite formation and maintains low resistance throughout cycling (Fig. 4f). Additionally, the crosslinked DAP-PETEA matrix not only generates a relatively homogeneous Li$^+$ flux, but also effectively eases the volume changes upon Li deposition, thus inhibiting any incipient dendrite growth[7]. Such a stable Li|HGPE interface with a low Li$^+$ diffusion energy barrier contributes to the superior performance of HGPE in Li metal batteries.

**Electrochemical performances evaluation**. Figure 5a, b show the charge-discharge profiles and rate performances of the Li| HGPE|KS6 graphite (sheet size of 4 μm, Supplementary Fig. 27) cell, respectively. During the charging process of Li|HGPE|KS6 graphite cells, three slopes at 4.21–4.50 V (stage III), 4.50–4.85 V

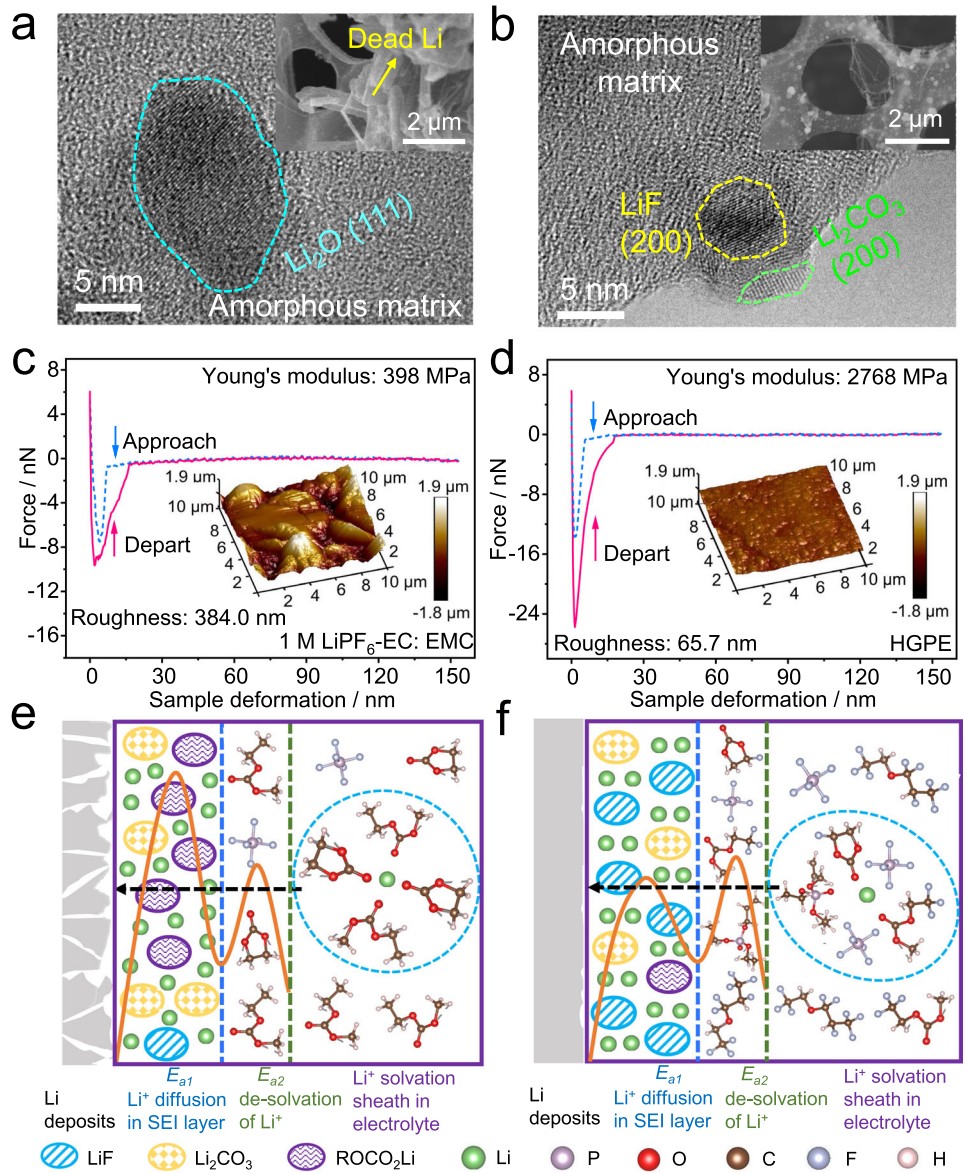

**Fig. 4 Experimental and theoretical investigations of the SEI on Li metal. a, b** TEM image of the SEI shell formed by repeatedly plating/stripping Li on Cu grids in **a** 1 M LiPF$_6$-EC: EMC electrolyte and **b** HGPE. **c, d** Force-displacement plots of **c** 1 M LiPF$_6$-EC: EMC derived SEI and **d** HGPE derived SEI. Corresponding three-dimensional atomic force microscope (3D-AFM) scanning images of SEI layers are shown in insets. **e, f** Schematics of the Li$^+$ deposition process in **e** 1 M LiPF$_6$-EC: EMC electrolyte and **f** HGPE. The aggregates in the light-blue dotted ovals represent the Li$^+$ solvation sheath in the bulk electrolyte. The curves correspond to the activation energies of $E_{a1}$ and $E_{a2}$. Scale bars: 5 nm in Fig. 4a, b; 2 μm in the insets of Fig. 4a, b.

(stage II), and 4.85–4.95 V (stage I) corresponded to the staged phase transition of graphite during anion insertion[9,41]. Subsequently, three plateaus associated with anions deintercalation from the graphite appeared in the discharge curves with potentials downshifted to 4.95–4.79 V, 4.79–4.40 V, and 4.40–4.0 V, respectively (Supplementary Note 3). This is in agreement with the CV curves in Supplementary Fig. 28 and the dQ/dV plot in Supplementary Fig. 29. The Li|HGPE|KS6 graphite cells delivered specific discharge capacities of 101.5, 99.8, 98.2, 95.8, 92.0, 88.3, and 81.0 mAh g$^{-1}$ at 0.1, 0.2, 0.5, 1, 2, 3, and 5 C (1 C = 100 mA g$^{-1}$ based on the mass of graphite), respectively (Fig. 5b). These are similar to those of the cells using 1 M LiPF$_6$-FEC: FEMC: HTE electrolytes, indicating that the gelation did not sacrifice the rate performance of batteries (Supplementary Fig. 30). When the C rate was switched back to 0.1 C, the capacity retention of the HGPE-based cell was 98.2% of the initial value, demonstrating that this battery system is highly robust and stable.

In contrast, the capacity of Li||KS6 graphite cell with 1 M LiPF$_6$-EC: EMC rapidly decreases to about 0 at 3 C, indicating a sluggish Li ions diffusion kinetics at the graphite|electrolyte interface.

Figure 5c shows the long-term cycling performance of Li||KS6 graphite DIBs employing various electrolytes at 1 C. The Li|1 M LiPF$_6$-EC: EMC|KS6 graphite cell exhibited an initial discharge capacity of 29.4 mAh g$^{-1}$, and the capacity suddenly dropped to 21.7 mAh g$^{-1}$ at the 169th cycle. This is probably caused by structure exfoliation and destruction of graphite originating from the co-intercalation of solvent molecules into the graphite interlayers, as well as the thickening of the CEI induced by the electrolyte oxidation[42]. The lifespan and reversible capacity of DIBs significantly increased with the adoption of fluorinated solvents (Supplementary Fig. 31). The Li||KS6 graphite cell using the HGPE demonstrated a high initial discharge capacity of 89.8 mAh g$^{-1}$ with a capacity retention of 93% after 1000 cycles, and the Coulombic efficiency was maintained at around 98.9%

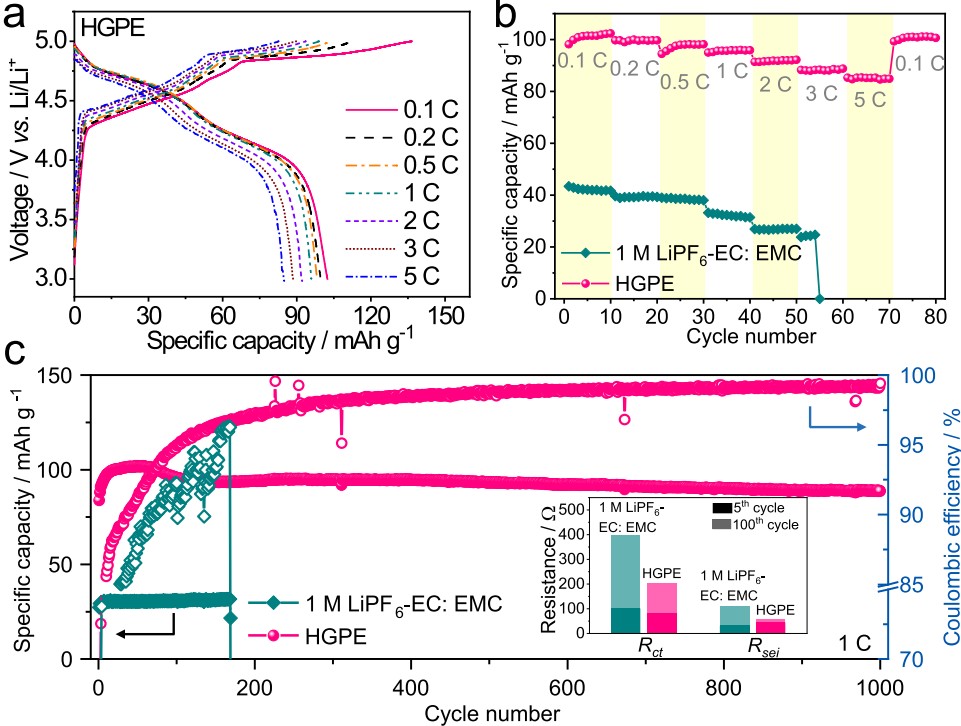

**Fig. 5 Electrochemical energy storage performance of the Li||KS6 graphite batteries. a** The charge-discharge voltage profiles and **b** Rate performances of a Li|HGPE|KS6 graphite cells employing 1 M LiPF$_6$-EC: EMC and HGPE under different C rates. **c** Long-term cycling performance Li||KS6 graphite cells at 1 C (1 C = 100 mA g$^{-1}$ based on the mass of graphite). The corresponding $R_{sei}$/$R_{ct}$ changes during cycling are shown in the inset.

except for the activation process in the first ten cycles (Fig. 5c). The above results were further corroborated by the small interfacial resistances ($R_{sei}$ and $R_{ct}$) of the HGPE-based cells, and the interfacial resistance changes were much smaller than in the cells using other electrolytes during cycling (inset of Fig. 5c, Supplementary Fig. 32, Supplementary Table 7, and Supplementary Note 4). This cycling stability is mainly because the HGPE effectively suppresses solvent co-intercalation and protects the structural integrity of graphite, thus allowing a highly reversible and durable insertion/extraction of anions into/from the KS6 graphite.

SRLMBs have been further developed by applying KS6 graphite as a conductive agent in the cathode of the LRO|HGPE|Li cells. As shown in Fig. 6a, during the charging of LRO|HGPE|Li and Li|HGPE|LRO/graphite cells, a sloping potential below 4.5 V corresponded to Li ion extraction from LRO cathode[43]. For the hybrid LRO/KS6 graphite cathode, an extra plateau at 4.9 V appears, which is ascribed to a "relay" intercalation step of PF$_6^-$ into the graphite. Supplementary Fig. 33 further validated that KS6 contributed about 6.2% of the areal capacity (i.e., 0.0505 mAh cm$^{-2}$) and 8.2% of the energy density (i.e., 3.2 Wh L$^{-1}$, Supplementary Note 5). Such a "shuttle-relay" process was highly reversible in the subsequent discharging. The SRLMB delivered a discharge capacity of 205.3 mAh g$^{-1}$, based on the total mass of the cathode active material and conductive agent, which was higher than that of the Li|HGPE|LRO cell with Super P as the cathode conductive agent (191.1 mAh g$^{-1}$, Fig. 6b). The cell can maintain a capacity of 188.0 mAh g$^{-1}$ after 100 cycles at 0.2 C (1 C = 250 mA g$^{-1}$ based on the mass of LRO) with a high capacity retention of 91.6 and 67.8% after 200 cycles at 0.5 C (Supplementary Fig. 34). This indicates that the PF$_6^-$ intercalation/deintercalation after Li$^+$ extraction/insertion can increase the battery capacity without sacrificing its cycling stability. In addition, the Li||LRO/graphite cell applying traditional 1 M LiPF$_6$-EC: EMC electrolyte suffered from a quick capacity fading

during cycling, demonstrating a poor electrode|electrolyte compatibility (Supplementary Fig. 35).

Single-layer SRLMB pouch cells with 50-μm-thick Li foil as anodes were assembled to further evaluate the battery performance under abuse conditions (Supplementary Fig. 36a). The Li| HGPE|LRO/graphite pouch cell not only showed adequate cycling performance (Supplementary Fig. 37), but also exhibited flexibility (i.e., consistently powering a red light-emitting diode (LED) under flatted, bent, or even clustered states (Fig. 6c, lower panels and Supplementary Movie 5). Whereas the cell using traditional liquid electrolyte losing power supply ability in the bent or clustered states (Fig. 6c, upper panels and Supplementary Movie 6). This verifies that the electrode|HGPE interfaces can maintain tight adhesion under significant shape deformations. Moreover, when aging the fully charged cells at 130 °C, a Li|| LRO/graphite pouch cell with 1 M LiPF$_6$-EC: EMC liquid electrolyte suffered from severe swelling and bulging due to the volatilization and thermal decomposition of the liquid electrolyte (Fig. 6d, inset), and the open circuit potential suddenly dropped to around 0 V at 1964 s, illustrating a contact failure inside the cell (Fig. 6d). In sharp contrast, owing to the high thermal stability of fluorinated solvents and leakage-free property of the gel, the shape and open-circuit voltage of Li||LRO/graphite pouch cell with HGPE did undergo not change at the 130 °C test (Fig. 6d). Meanwhile, the temperature excursion of a fully charged Li| HGPE|LRO/graphite pouch cell was lower than that of a Li|1 M LiPF$_6$-EC: EMC|LRO/graphite pouch cell during the nail penetration safety tests (Supplementary Fig. 36b). All these enable a highly safe operation of SRLMBs in practical applications. Considering that graphitic carbon materials are not only served as conductive agents in cathodes, but also widely used as coating layers on cathode materials and/or Al current collector. In this work, we demonstrated that the "shuttle-relay" concept utilizing such graphitic carbon components in the cathode can

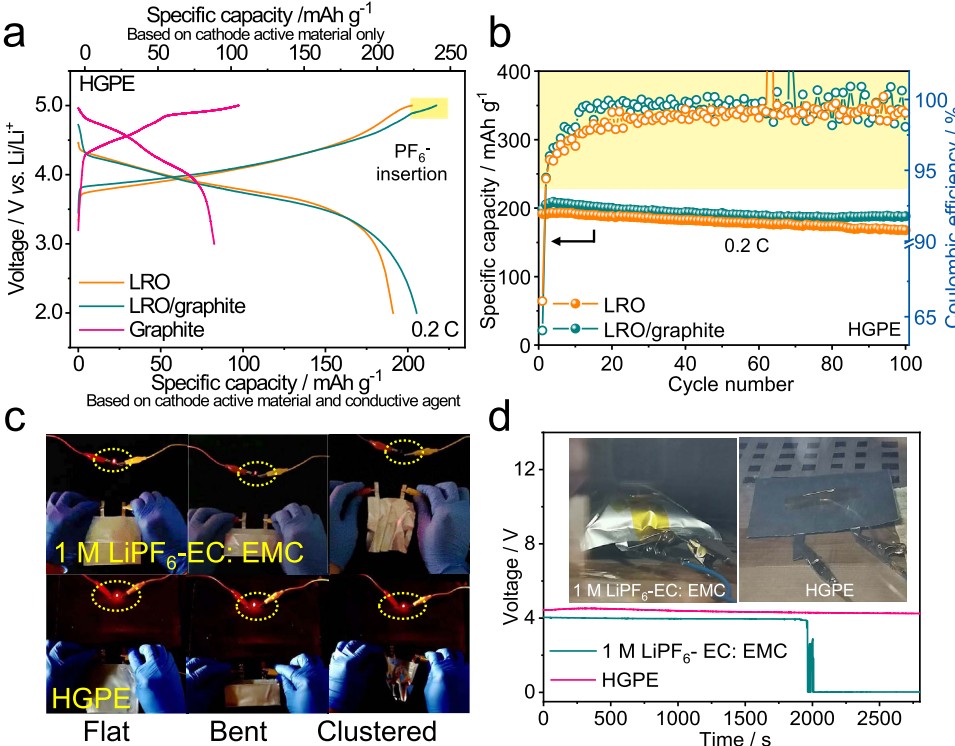

**Fig. 6 Electrochemical energy storage performance of the SRLMBs. a** Charge-discharge curves of the Li|HGPE|KS6 graphite, Li|HGPE|LRO, and Li|HGPE|LRO/graphite hybrid cells at 0.2 C. **b** Cyclic performances of Li|HGPE|LRO and Li|HGPE|LRO/graphite hybrid cells at 0.2 C. **c** Optical images of red LEDs powered by Li|1 M LiPF$_6$-EC: EMC|LRO/graphite and Li|HGPE|LRO/graphite pouch cells. **d** Open circuit voltage changes of fully charged Li|1 M LiPF$_6$-EC: EMC|LRO/graphite and Li|HGPE|LRO/graphite pouch cells at 130 °C during the aging time. Optical images of the pouch cells after aging at 130 °C for half an hour are shown in the inset.

provide additional capacity, which increases the energy density of existing Li batteries[44].

**Electrochemical mechanism of PF$_6^-$ intercalation/deintercalation in the HGPE.** In situ X-ray diffraction (XRD) tests were conducted to further investigate the operation mechanism of PF$_6^-$ intercalation/deintercalation in the presence of different electrolytes at 0.05 C. The in situ XRD patterns and charge/discharge curves during the initial cycle are shown in Supplementary Fig. 38, and the corresponding intensity contour maps are presented in Fig. 7a, b, respectively. In the Li||KS6 graphite cells using traditional 1 M LiPF$_6$-EC: EMC liquid electrolyte, the graphite (002) diffraction peak gradually shifted from 26.6° to 24.1° during the charging process. The corresponding interlayer spacing (*d*) values of graphite can be calculated from the XRD pattern according to the Bragg equation[45]:

$$d = \lambda / (2 \sin \theta) \qquad (2)$$

where $\theta$ is the diffraction angle between the incident X-rays and the corresponding crystal plane, and $\lambda$ is the X-ray wavelength (i.e., 0.15406 nm). The increase of graphite $d_{(002)}$ interplanar spacing from 0.335 nm at 3.0 V to 0.370 nm at 5.0 V is consistent with the intercalation of the PF$_6^-$ anions into the graphite interlayer (Supplementary Fig. 39). In the subsequent discharge process, however, no distinct position change of the graphite (002) peak was observed, indicating a constant graphite $d_{(002)}$ spacing caused by the blocked PF$_6^-$ anion stripping from the graphite host (Fig. 7a). This has been further confirmed by the TEM image of the graphite cathode after cycling, in which the lattice spacing from XRD (0.370 nm) is well-consistent with the calculated value from the TEM (Fig. 7c, inset). The thickness of the CEI derived from the oxidative decomposition of carbonate

solvents is as high as 7.8 nm in 1 M LiPF$_6$-EC: EMC liquid electrolyte, which strongly hinders the PF$_6^-$ stripping and causes the irreversibility during cycling (Fig. 7c). In sharp contrast, in the Li|HGPE|graphite cell, the graphite (002) diffraction peak gradually shifted to 24.1° (i.e., interlayer spacing of 0.370 nm) during the charge process, and reversibly reverted to 26.50° (i.e., interlayer spacing of 0.336 nm) when discharged back to 3.0 V (Fig. 7b). The TEM image of the cycled graphite cathode exhibited lattice stripes with a spacing of 0.336 nm (Fig. 7d, inset), which is in accordance with the in situ XRD results and almost the same as that of the pristine graphite powder (0.335 nm, Supplementary Fig. 40). This validates a highly reversible PF$_6^-$ intercalation into/deintercalation from the graphite without structural deterioration. Moreover, the thickness of the CEI formed in the HGPE is only 1.4 nm, indicating a suppressed electrolyte oxidation with reduced interfacial resistance.

Ex situ postmortem in-depth XPS measurements were performed on graphite cathodes cycled in various electrolytes to further analyze the components of CEI layers. For the Li|1 M LiPF$_6$-EC: EMC|graphite cell, peaks at about 532 eV (C=O), 530 eV (ROCO$_2^-$), and 528 eV (Li$_2$O) in O 1*s* spectrum and 687.5 eV (PF$_6^-$), 686 eV (Li$_x$PO$_y$F$_z$), and 685 eV (LiF) in F 1*s* spectrum appeared on the cycled cathode surface (a sputtering time of 0 s, Fig. 7e)[9,20,32], which is in agreement with Li 1*s* spectrum in Supplementary Fig. 41. In addition, the peaks at about 136, 134, and 131 eV in the P 2*p* spectrum are related to PF$_6^-$, Li$_x$PO$_y$F$_z$, and P-C/P-O-C[46], respectively (Supplementary Fig. 42). These results suggest that in 1 M LiPF$_6$-EC: EMC liquid electrolyte, the CEI on the graphite cathode is mainly composed of abundant alkyl carbonate (e.g., ROLi) and polycarbonate as oxidation products of carbonate solvents (as further verified in the C 1*s* spectrum in Supplementary Fig. 43, and Li$_x$PO$_y$F$_z$ and

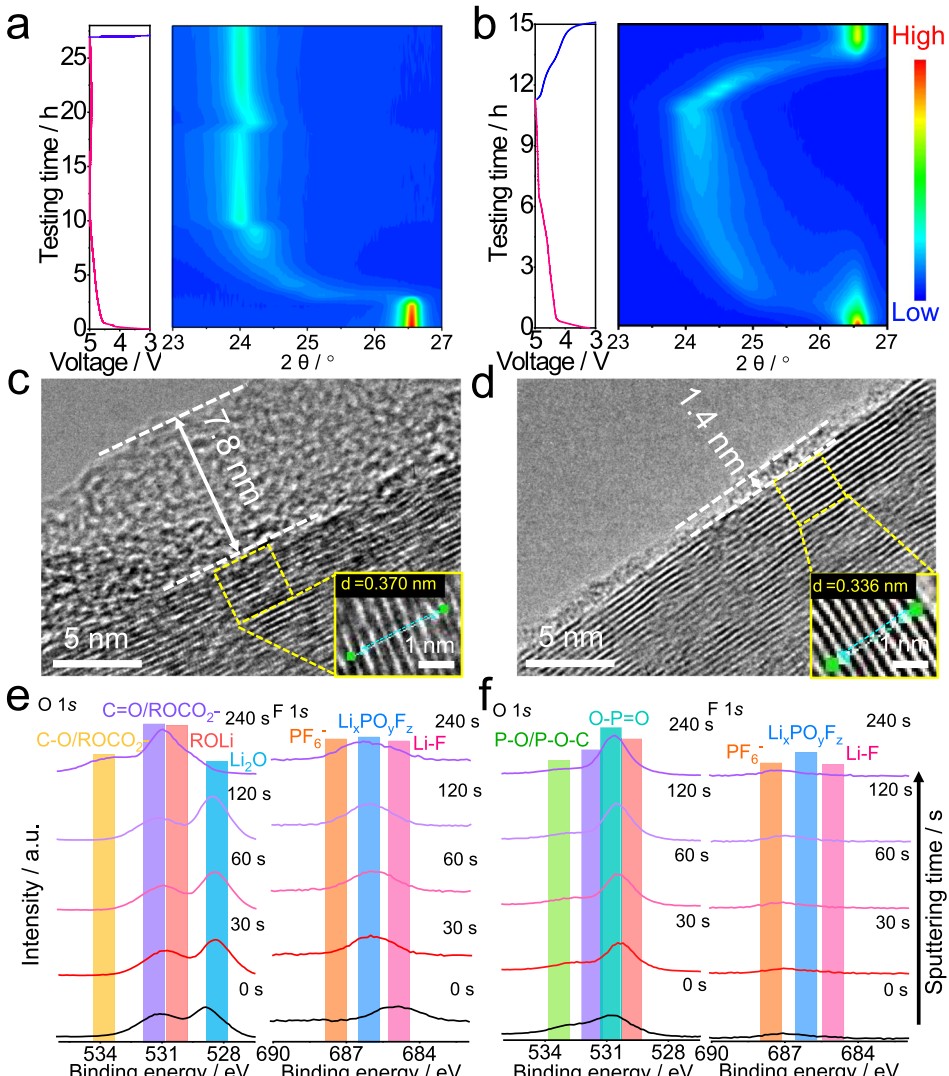

**Fig. 7 Electrochemical mechanism of PF$_6^-$ intercalation/deintercalation. a, b** Intensity contour maps obtained from the in situ XRD patterns of **a** Li|1 M LiPF$_6$-EC: EMC | graphite and **b** Li|HGPE | graphite cells at 0.05 C. **c, d** TEM images of the KS6 graphite cathodes from **c** Li|1 M LiPF$_6$-EC: EMC | KS6 graphite and **d** Li|HGPE | KS6 graphite cells after one cycle at 0.05 C. **e, f** O 1$s$ and F 1$s$ XPS in-depth spectra of the KS6 graphite cathodes obtained from **e** Li|1 M LiPF$_6$-EC: EMC | KS6 graphite and **f** Li|HGPE | KS6 graphite cells after one cycle at 0.05 C. Scale bars: 5 nm in Fig. 7c, d; 1 nm in the inset of Fig. 7c, d.

LiF originated from the decomposition of LiPF$_6$ in Fig. 7e). More importantly, a large amount of intercalated PF$_6^-$ anions remained in the graphite interlayers, and the intensity of ROCO$_2^-$ from carbonates abruptly increased at a depth of 20 nm (i.e., a sputtering time of 240 s). These demonstrate the inhibited stripping of PF$_6^-$ anions from the graphite host and the co-intercalation of solvent molecules, which causes the irreversible capacity loss of the cells at 0.05 C.

In contrast, for the surface of the graphite cathode from the cycled HGPE-based cell, two new O 1$s$ peaks at about 533 and 531 eV in Fig. 7f are assigned to P-O-C and O-P=O as oxidative decomposition products of DAP[47,48], which is consistent with the P 2$p$ spectra in Supplementary Fig. 42. Additionally, the peak intensities of PF$_6^-$, Li$_x$PO$_y$F$_z$, and LiF in F 1$s$ spectrum and the ROCO$_2^-$ from the C 1$s$ spectrum significantly decreased (Fig. 7f and Supplementary Fig. 43). The above results suggest that phosphorus-containing substances in the CEI (i.e., P-O-C and O-P=O) can suppress the decomposition of electrolytes and form a thin CEI to ensure a reversible PF$_6^-$ deintercalation (Figs. 5c, 6a, and 7d). Considering allyl groups can easily undergo polymerization[47], such a CEI film may originate from

polyphosphoesters generated by the electropolymerization of residual DAP monomer on the carbon-oxygen rich graphite surface (Supplementary Fig. 44). Moreover, in the XPS depth profiles of the Li|HGPE | graphite cell, no noticeable peaks were observed from the carbonate solvents or their decomposition products. This confirms that solvent molecule co-intercalation can be effectively inhibited by such protective CEI, which preserves the cathode against structure destruction and facilitates the superior electrochemical performance of the HGPE-based DIBs and SRLMBs.

In conclusion, we showcased an HGPE facilitating highly reversible insertion/extraction of PF$_6^-$ anion into/from graphite interlayers. The HGPE prepared via a facile in situ thermally initiated polymerization possesses high ionic conductivity (1.99 mS cm$^{-1}$) and safety (i.e., nonflammable and free of liquid leakage). The synergistic effect of fluorinated solvents, polymer matrix, and the residual DAP monomer in the HGPE contributes to stable electrode|HGPE interfaces, thus endowing oxidative stability up to 5.5 V vs. Li/Li$^+$, high Li deposition/stripping Coulombic efficiency of 99.7%, and appealing cycling stability of graphite cathodes with 93% capacity retention after 1000 cycles.

Utilizing this HGPE, as a proof-of-concept, we developed a quasi-solid-state SRLMB with a hybrid LRO cathode by applying KS6 graphite as the conductive agent, in which a reversible insertion of $PF_6^-$ anions into the graphite occurred after the stripping of Li ions from the LRO. By unlocking the anion capacity contribution and elaborately modifying interfacial compatibility, this hybrid design exhibits significant merits in terms of overall energy density and cycling stability, which can be extended to other conventional cathode materials.

## Methods

**Preparation of the HGPE**. LiPF$_6$ (CAPCHEM, 99.99%), DAP (Macklin, 96.0%), PETEA (98%, Sigma-Aldrich), and 2, 2′-azobis (2-methylpropionitrile) (AIBN, Aladdin, 99%) were sealed and stored at −20 °C before use to protect them from deterioration. EC (DoDoChem, 99.98%), EMC (DoDoChem, 99.9%), FEC (DoDoChem, 99.95%), FEMC (DoDoChem, 99.95%), HTE (J&K Scientific Ltd, China), and dimethyl carbonate (DMC, DoDoChem, 99.95%) were used without further purification. To prepare the HGPE, 3 wt% DAP monomer, 1.5 wt% PETEA crosslinker, and 0.1 wt% AIBN initiator were co-dissolved in a liquid electrolyte consisting of 1 M LiPF$_6$ in a nonaqueous mixture of FEC: FEMC: HTE (1: 6: 3 by volume) to form a precursor solution. Then, 1 g precursor solution was sealed in glass bottles. The above processes were in an Ar-filled glove box under atmospheric pressure. The precursor solution was further heated by a vacuum oven at 70 °C for half an hour to obtain translucent HGPE. The contents of trace water in the LiPF$_6$-based liquid electrolytes were detected to be around 10–15 ppm by the Karl Fischer method (831 KF Coulometer, Metrohm). The polymer matrix of HGPEs was separated and purified for further characterization as follows: the as-obtained HGPE was firstly mashed into pieces and washed with acetone. Subsequently, the mixture was centrifuged at 10,000 rpm for 15 min to separate the white precipitates. The above procedures were repeated three times. After vacuum drying at 120 °C, the as-obtained precipitates were dialyzed against deionized water for 3 days to further remove the residual ions. Then the precipitates were vacuum-dried at 120 °C to obtain the separated polymer matrix.

**Characterization of HGPE**. The conversion rate of monomers was measured by $^1$H NMR analysis (Bruker AVANCEIII400) with dimethyl sulfoxide-d6 as a solvent, which could be estimated from the integrated area ratio of CH$_2$= on the monomers in the polymerized gel/solution to that in the pristine precursor solution. The CH$_2$- on FEMC solvent was set as a reference. FTIR spectra of the DAP monomer, PETEA crosslinker, and the polymer matrix of HGPE were measured with a Bruker Vertex70 instrument. Raman spectroscopy was conducted by a Micro-laser confocal Raman spectrometer (Horiba LabRAM HR800, France) at room temperature with a 532 nm laser. In the combustion test, 1 g electrolyte samples were poured into a dish, and then optical photographs and movies were recorded after the samples were ignited. The weight loss of electrolyte samples as a function of aging time was measured in an open environment at 60 °C. The ionic conductivities of the electrolyte samples were measured by EIS at an alternating potential amplitude of 5 mV and six points per decade with a frequency range of $10^5$ to 1 Hz on a VMP3 multichannel electrochemical station (BioLogic Science Instruments, France). The test cells were assembled by immersing two stainless steel blocking electrodes into electrolyte samples. Before the conductivity measurements, the test cells were maintained at each test temperature (from 0 to 90 °C) for at least 30 min to reach thermal equilibrium. The Li ion transference number ($t_{Li}^+$) of the electrolyte samples was tested using the method described by Abraham et al[49]. The processes were as follows: symmetric Li|HGPE|Li cell was assembled and then the polarization currents, including the initial ($I^o$) and steady-state ($I^{ss}$) current values, were recorded under a small polarization potential ($\Delta V$) at 10 mV. Simultaneously, the initial and steady-state values of the bulk resistances ($R_b^o$ and $R_b^{ss}$) and electrode|electrolyte interfacial resistances ($R_i^o$ and $R_i^{ss}$) were examined via EISs before and after the potentiostatic polarization. The $t_{Li}^+$ was calculated based on the following equation:

$$t_{Li}^+ = \frac{I^{ss}(\Delta V - I^o R_i^o)}{I^o(\Delta V - I^{ss} R_i^{ss})}. \tag{3}$$

The electrochemical stabilities of the electrolytes were studied by LSV tests on a three-electrode system at a scanning rate of 5 mV s$^{-1}$ using the VMP3 electrochemical station. Platinum foil was used as the working electrode, while Li foil was used as the counter and the reference electrodes in this system. The oxidation potential values of electrolytes were recorded as the voltage when the current increased to 10 μA.

To evaluate the compatibility of electrolytes with Li metal, galvanostatic cycling measurements consisting of repeated 2 h charge–2 h discharge cycles were carried out in symmetrical Li||Li cells at 0.5 mA cm$^{-2}$. The EIS measurements were performed at an alternating potential amplitude of 5 mV and recorded six points per decade with a frequency range of $10^5$ to 1 Hz on a VMP3 on the VMP3 multichannel electrochemical station. Symmetric Li||Li cells using various electrolytes were cycled 20 times at a current density of 0.5 mA cm$^{-1}$ for activation energy measurements. Then the cycled cells were kept under 283, 293, 303, 313,

and 323 K to record the temperature-dependent EISs. The SEI resistance ($R_{sei}$) and ion transfer resistance ($R_{ct}$) values were obtained by fitting the EISs via an equivalent circuit shown in the inset of Supplementary Fig. 16a. Then the activation energy ($E_a$) was derived from the Arrhenius equation as follows:

$$k = \frac{T}{R_{res}} = A \exp\left(-\frac{E_a}{R}\right) \tag{4}$$

where $k$ represents the rate constant, $T$ is the absolute temperature, $R_{res}$ represents $R_{ct}$ or $R_{sei}$, A is the preexponential constant, $E_a$ is the activation energy, and R is the standard gas constant[50,51].

The Coulombic efficiencies of Li depositing/stripping in different electrolytes were investigated in Li||Cu coin cells, using the method reported by Zhang et al[36]. The Cu substrate was preconditioned with one Li deposition/stripping cycle with a capacity of 5 mAh cm$^{-2}$ at 0.5 mA cm$^{-2}$. After depositing 5 mAh cm$^{-2}$ Li reservoir ($Q_T$) on the Cu substrate at 0.5 mA cm$^{-2}$, the cell was charged-discharged with a capacity of 1 mAh cm$^{-2}$ ($Q_C$) for $n$ cycles, followed by a final exhaustive stripping of the remaining Li reservoir to 1 V at 0.5 mA cm$^{-2}$. The final stripping charge ($Q_S$), corresponding to the quantity of Li remaining after cycling, was measured. The average CE over $n$ cycles can be calculated as[36]

$$CE_{avg} = \frac{nQ_C + Q_S}{nQ_C + Q_T}. \tag{5}$$

The Cu substrates were harvested from dissembled Li||Cu cells after deposition of 1 mAh cm$^{-2}$ Li at 0.2 mA cm$^{-2}$ for further FE-SEM (SU8010) characterization. 1 mAh cm$^{-2}$ Li was repeatedly plated on and stripped off a Cu grid for ten cycles at 0.2 mA cm$^{-2}$ to obtain a vacant SEI shell for TEM (Tecnai G$^2$ F30). The Cu foils after plating-stripping 1 mAh cm$^{-2}$ Li for ten cycles at 0.2 mA cm$^{-2}$ were subjected to in-depth XPS (PHI 5000 VersaProbe II, in which the thickness values in the XPS depth profiles were estimated from the calibrated sputtering of SiO$_2$) and AFM (Bruker Dimension Icon) characterizations. All above electrochemical energy storage tests were carried out in an environmental chamber at 25 °C and the error of the temperature measurements was no more than 1 °C.

**Battery assembly and characterization**. The graphite cathode, LRO cathode, and LRO/graphite hybrid electrode were prepared by a slurry-coating method without calendaring step. To obtain KS6 graphite electrode, a slurry mixture consisting of 70 wt% conductive graphite (KS6, Canrd Co. Ltd.), 20 wt% carbon nanotubes (CNT) as a conductive agent, and 10 wt% polyvinylidene fluoride (PVDF, Macklin, AR 90%) as a binder in anhydrous N-methyl-2-pyrrolidone (NMP, Sigma-Aldrich) was cast onto a carbon-coated aluminum (Al) foil and then dried at 120 °C overnight under vacuum. The LRO-based electrodes were prepared following similar methods with an LRO: Super P: PVDF weight ratio of 80:10:10 and LRO: KS6: CNT (applying to enhance the electrode electronic conductivity in Supplementary Fig. 45): PVDF weight ratio of 80: 8: 2:10, respectively. The mass loadings of the active materials on the KS6 graphite electrode and LRO/KS6 graphite electrode were around 1.0 and 3 mg cm$^{-2}$, respectively. CR2032 coin cells were assembled in an Ar-filled glove box using Celgard 2400 separator and Li metal anodes. Precursor solution containing 3 wt% DAP, 1.5 wt% PETEA, and 0.1 wt% AIBN dissolved in 1 M LiPF$_6$-FEC: FEMC: HTE (1: 6: 3 by volume) electrolyte was injected into the separator and filled into the cells. The electrolyte/graphite ratio in each cell was uniformly set at about 60 μL mg$^{-1}$. Then the assembled cells were aged at room temperature for 2 h to ensure the precursor solution wetted the electrodes sufficiently. Subsequently, the cells were heated at 70 °C for 1 h in a vacuum oven to ensure an in situ copolymerization of DAP monomer and PETEA crosslinker to get the HGPE-based cells.

**Assembly of soft-packing SRLMBs**. A Li|HGPE|LRO/graphite hybrid pouch cells were assembled in a glove box. The compositions of electrodes and precursor solution were the same as the above-mentioned coin cells. Nickel and Al strips were joined anchored to the side of anode and cathode as the electrode tabs, respectively. The electrodes (the size of cathode: 56 mm × 43 mm and anode: 58 mm × 45 mm) and separator were laminated together to construct the battery core and assembled into Al-plastic film packages, followed by injecting the precursor solution (around 250 μL) into the packages and sealing them under vacuum. The size of the pouch cell is 75 mm × 55 mm × 0.5 mm and its volume is around 2.06 cm$^3$. Subsequently, the assembled cells were aged at room temperature for 6 h to ensure the precursor solution was well-wetted into the electrodes, and then heated at 70 °C to gelate the precursor solution. Finally, the cells were aged at 25° C for 12 h, and degassed after the initial cycle. The assembled HGPE-based cells were cycled at between 2–5 V at 0.2 C. The pouch cell was charged to 5 V and held at the charge cutoff voltage for 1 h before the safety tests. The nail penetration test of pouch cells was conducted by Battery Nail Puncture Tester (DAMS DMS-9982) at a depth of 130 mm with a piercing speed of 25 mm s$^{-1}$. The high-temperature stability test of electrolytes was performed by detecting the voltage change of the pouch cells at 130 °C in half an hour with High-temperature Isolation Equipment (DAMS DMS-9987).

The as-developed Li||graphite cells were charged-discharged between 3.0 and 5.0 V and Li||LRO cells were charged-discharged in a voltage range of 2.0 to 5.0 V on a Land 2001 A battery testing system at 25 °C. After designated cycling tests, the cells were dissembled in an Ar-filled glove box and repeatedly rinsed with 1 mL DMC before postmortem analysis. The air-sensitive electrode samples were transferred into the

vacuum chambers for in-depth XPS at an inert atmosphere. Samples for TEM tests were exposed to air for no more than 5 s before being transferred to the vacuum chamber. The dQ/dV curves were calculated from the discharge/charge profiles with a set voltage interval of 10 mV. CVs of the assembled DIBs were tested using the VMP3 electrochemical working station at a scanning rate of 0.5 mV s$^{-1}$, while EISs were examined in a frequency of 10$^{-2}$ to 10$^5$ Hz with six points per decade by applying a potential amplitude of 5 mV. For the in situ XRD experiments, Li‖graphite Swagelok cells were assembled applying beryllium foils as both cathode current collector and X-ray window. The graphite cathodes herein were composed of 90 wt% KS6 graphite and 10 wt% PVDF to exclude the effect of a conductive agent in XRD patterns. The in situ XRD patterns were characterized on a Rigaku D max 2500 diffractometer with Cu Kα radiation (λ = 1.5418 Å).

**Theoretical calculations**. All the spin-polarized calculations were performed using a Vienna ab initio simulation package (VASP), which was a plane-wave density functional code. The electron-electron exchange and correlation inter-actions were described by using the generalized gradient approximation (GGA) with the Perdew–Burke–Ernzerhof (PBE) functional form. The projector augmented-wave (PAW) method was employed to describe the interaction between the core and valence electrons. To better describe the interactions between molecules, van der Waals (vdw) interactions were included by the DFT-D3 method of Grimme.

## Data availability

The data that support the findings of this study are available from the corresponding author upon reasonable request.

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

## Acknowledgements

Prof. B. Li would like to acknowledge the support funded by the National Nature Science Foundation of China (No. 51872157), Shenzhen Key Laboratory (ZDSYS201707271615073),

and Guangdong Technical Plan Project (No. 2017B090907005). Prof. G. Wang would like to acknowledge the support from Australian Research Council (ARC) Discovery Projects (DP200101249 and DP210101389) and the ARC Research Hub for Integrated Energy Storage Solutions (IH180100020). We would like to thank Prof. Atsuo Yamada from The University of Tokyo for giving valuable advice on this work.

## Author contributions

D.Z., M.A., B.L., and G.W. conceived and designed this work. D.Z. and J.W. performed the experiments and wrote the manuscript. S.W. conducted the AFM characterizations. D.S., X.W., Q.L., F.K., and T.R. discussed the results and participated in the preparation of the paper.

## Competing interests

The authors declare no competing interests.
