## [Peer Review File · Nature Communications]

REVIEWER COMMENTS

Reviewer #1 (Remarks to the Author):

Dear Authors,

the manuscript titled "A "shuttle-relay" lithium metal battery enabled by heteroatom-based gel polymer electrolyte" is dealing with the preparation of GPE based LMBs in which a LRO based cathode is used with lithium metal anode. The innovative concept of combining the DIB and rocking chair concept is promising for various applications. GPEs proposed by the Authors demonstrated excellent resistance towards Li-dendrite growth and a good relation to their stability to the type of interphase using various experimental methods and theoretical calculation. However, there are lots of questions to be answered when the HGPE is considered along with the overall cell concept and its applicability. Some of the main comments are given below:

1. From a practical point of view, Authors have claimed that there could be a 7 % increase in the overall cell capacity. However, Authors have used 12% of carbon content in the electrode. However, practically, the carbon content in an cathode is mostly below 5 wt%. so in such a case, how will their innovation be useful if one replaces this 7-8 % of carbon with the active material?. Thus, achieving an increase in the specific capacity will be challenging if not impossible or detrimental. Secondly, the stability of LRO above 4.70 V, which is a huge challenge especially during long term cycling. May be for achieving 7% extra capacity, long-term cyclability might be risked. Besides, the volume change in the graphite in the cathode could generate contact issues in the long run. Indeed, the coulombic efficiency is fluctuating in Fig. 5E for the HGPE/LRO-graphite based cell. Why did the Authors used separate conductive additives in the electrodes, their role in the performance is not well documented. Must be compared to understand a clear picture.

2 Serious concerns are related to the HGPEs. Authors showed the polymer matrix as a Copolymer network. However, they have used a highly reactive acrylate and a very low reactive allylic phosphate and tried to copolymerize it together, expecting it to be a co-polymer network as demonstrated in the scheme. Due to the lower reactivity, allyl will polymerize slowly and separately. The acrylates due to their high reactivity, will polymerize very fast. This is also reflected in the results, for examples, they obtained a very low conversion rate of 71%. Acrylates as unreacted monomers will create issues with the lithium metal in a full cell. From polymerization point of view, Authors must sort out this polymerization issue and identify the polymer matrix structure in details. Authors should study the kinetics of acrylate and DAP polymerization separately. Indeed the polymerization time of 1 hour is not sufficient for DAP to form a polymer of significant molecular weight. In the best case scenario Authors may achieve sIPN/IPNs as the polymer matrix. Then the use of LiPF₆ and polymerization temperature are challenging combinations. Especially, the polymerization temp of 70°C and PF₆ stability are a serious issue. Indeed, at that temperature, acrylates might be polymerized by cationic polymerization induced by PF₆ anions, which decompose into HF and PF₅. Indeed Authors didn't purify the monomers and used as it is, which will induce water molecules in the electrolyte and form various unwanted products. These issues are serious from the polymerization front. Also, it is not clear why the Authors used vacuum for polymerization and how are they going to control the evaporation of the solvents at 70°C and vacuum. Such issues will create bubbles and uneven contacts during the in situ processing. The AIBN decomposition is also different in vacuum, which will create further issues for the polymerization.

3. PF₆ interrelation leads to a change in PF₆ anion concentration in the electrolyte. How this issue is going to affect the ionic conductivity, and overall voltage of the cell during the initial and final stages of charge /discharge processes? Did the Authors study the influence of salt concentration in the electrolyte?

4. From the manuscript one can understand that Authors have used a Li/SS cell for ECSW. They used a wide scan range. But, there are already passivation and other decomposition happening at the electrodes during the reduction process. For example, SS is involved in a reaction around -

0.43 V according to the Authors. Then, how can we consider this electrode as a working electrode (inert working electrode) while studying the anodic stability of a GPE?. Indeed, the polymer is also not the same after the reduction process as mentioned by the authors, the monomers, PF6, and solvents decompose to form SEI layer in the reduction stage. Besides, in figure 8, supporting information, Authors have randomly selected the oxidation stability values. It is not clear what is the standard limiting value they have used to determine the electrochemical stability. Also in most cases, only passivation is observed not a decomposition, thus while assigning stability values, one must be extra careful. Besides, the membranes are already applied for reduction in the same cell. Thus, the discussion on ECSW is superficial. Thorough analysis is needed.

5. Authors must check for grammatical errors and spelling errors throughout the manuscript, for example: stripping vs. strapping. Then in the abstract, first sentence is not very clear.

6. page: 4, line 73, construct a high-resistance cathode electrolyte interface (CEI) on the cathode surface, is it an interface or interphase?

7. Figure 2 f caption is wrong

8. Figure 2 b, FTIR curves must be provided before and after polymerization, otherwise readers cannot understand the difference.

9. Fig 23, CV peak behavior after the 5th cycle must be thoroughly discussed. currently, main peaks occurring after the cycling is mentioned, but not explained the reasons behind such changes.

Due to these reservations and grave concerns about the polymer matrix structure and applicability, my recommendation is major revisions.

best regards,

Reviewer #2 (Remarks to the Author):

In this work, the authors have demonstrated a high-performing "shuttle-relay" concept battery where capacity of graphite as a stand-alone or as an additive cathode material under high-voltages is reversibly utilized. The electrolyte utilized has been thoroughly characterized as well as its interaction with the anode and cathode during cycling. While the solutions employed here for a high-functioning "shuttle-relay" cell have individually been demonstrated 'in-essence', for example, the use of fluorinated solvents to get dense lithium metal deposition, use of fluorinated solvents with or without CEI-forming additives to get reversible capacity from graphite cathode, utilizing concentrated-electrolyte concept, etc, this work should be published in Nature Communications for its detailed characterizations with minor revision. Below are some comments for the authors:

1. Have the authors quantified the amount of residual DAP in their HGPE? It would be important to know as this residual DAP seems to be helping significantly in forming an effective SEI and CEI.
2. Throughout the paper, the distinguishing cause behind some improvement of performance is almost always the residual DAP decomposition to improve SEI/CEI. What do authors think about putting DAP as an electrolyte additive and testing just the liquid electrolyte?
3. Page 32, line 570, the authors wrote: "The HGPE prepared via a facile in-situ.....and safety (i.e. non-flammable and free of liquid leakage)." I believe that the point "free of liquid leakage" has not been satisfactorily demonstrated especially during cycling. The data on pouch cell swelling in Fig 5g is not enough evidence of non-leakage as that might have just to do with low volatility of fluorinated solvents. Even the weight loss data in Supplementary Fig. 5 is not a direct evidence of leakage prevention during cycling. The authors can consider pursuing this further.
4. Page 22, line 405, the authors wrote: "These are much higher than those of cells using 1 M LiPF6-FEC: FEMC and 1 M LiPF6-FEC: FEMC: HTE electrolytes (Supplementary Fig. 25)." I disagree

that the performance of Li|HGPE|KS6 cell is significantly better than that of 1MLiPF6-FEC:FEMC:THE liquid electrolyte shown in SI Fig 25. The performance of these two are actually quite similar. And especially since the cathode loadings are very low (1 mg/cm²), some error in the weight measurements can cause the amount of difference in capacity seen between HGPE and the fluorinated liquid mixture shown in SI Fig 25, 26. Please rephrase this sentence if makes sense.

0. Page 18, line 340, there is a typing error. "... (Supplementary Fig. 18, insets)" should be "Supplementary Fig. 19".

1. Page 18, line 337, please confirm whether the XPS analysis was performed on the cycled TEM grids or not.

2. In the caption of Supplementary Fig. 5, the readers are directed to Supplementary Note 3. The Note 3 is unrelated. Please check.

Dear Dr. Rinaldo Raccichini,

Thank you very much for considering our manuscript entitled “A “shuttle-relay” lithium metal battery enabled by heteroatom-based gel polymer electrolyte” that has been submitted to **Nature Communications**. It is an honor for us that the editor and reviewers have provided very detailed comments and professional revision suggestions. Herein, we present a point-by-point response to the reviewer’s comments. All revisions are highlighted by yellow background and underline in the revised manuscript. We hope that this revised manuscript could meet the requirements for publication on your prestigious journal.

Thank you very much for your considerations. We are looking forward to hearing from you.

Best regards,

Guoxiu Wang on behalf of all co-corresponding authors and co-authors.

Response letter

Reviewer #1:

The manuscript titled A “shuttle-relay” lithium metal battery enabled by heteroatom-based gel polymer electrolyte` is dealing with the preparation of GPE based LMBs in which a LRO based cathode is used with lithium metal anode. The innovative concept of combining the DIB and rocking chair concept is promising for various applications. GPEs proposed by the Authors demonstrated excellent resistance towards Li-dendrite growth and a good relation to their stability to the type of interphase using various experimental methods and theoretical calculation. However, there are lots of questions to be answered when the HGPE is considered along with the overall cell concept and its applicability. Some of the main comments are given below:

1) From a practical point of view, Authors have claimed that there could be a 7 % increase in the overall cell capacity. However, Authors have used 12% of carbon content in the electrode. However, practically, the carbon content in a cathode is mostly below 5 wt%, so in such a case, how will their innovation be useful if one replaces this 7-8 % of carbon with the active material? Thus, achieving an increase in the specific capacity will be challenging if not impossible or detrimental. Secondly, the stability of LRO above 4.70 V, which is a huge challenge especially during long term cycling. May be for achieving 7% extra capacity, long-term cyclability might be risked. Besides, the volume change in the graphite in the cathode could generate contact issues in the long run. Indeed, the coulombic efficiency is fluctuating in Fig. 5E for the HGPE/LRO-graphite based cell. Why did the Authors used separate conductive additives in the electrodes, their role in the performance is not well documented? Must be compared to understand a clear picture.

Response: We appreciate the reviewer’s valuable comments that help to improve the quality of our work. A point-to-point response to this comment is described below:

1) As commented by the reviewer, we did use 10 wt% of carbon (including conductive graphite and carbon nanotube (CNT) as conductive agent in the cathode, which is higher than that in the commercial cells (<5 wt%). We acknowledge that in the commercial batteries, the additional capacity contribution originating from the PF₆ insertion into conductive graphite may be limited. However, graphitic carbon materials are not only served as conductive agents in cathodes, but also widely used as coating layers on cathode materials^[1] and/or Al current collector^[2]. Therefore, the content of graphitic carbon materials is considerable in the cathodes of commercial cells. In this work, we demonstrated that the “shuttle-relay” concept utilizing such graphitic carbon component in the cathode can provide additional capacity, which increases the specific energy of existing lithium batteries. We have added this explanation on Page 24 in the revised Manuscript.

2) The reviewer thought the long-term cyclability of the “shuttle-relay” Li metal battery (SRLMBs) might be risky due to the 4.7 V operating voltage of LRO and the repeated volumetric change of graphite, which caused fluctuations of Coulombic efficiency during cycling. In our work, we optimized electrolyte composition to achieve

a high electrochemical stability up to 5.8 V vs. Li/Li⁺, alleviating the electrolyte oxidation at high voltage. In addition, it should be noted that the Li ions stripping from LRO is a volumetric shrinkage process, while the subsequent intercalation of PF₆⁻ anion into graphite is volumetrically increment. Therefore, the total volume change for the cathode is actually decreased in the SRLMBs, which facilitates the structural integrity of cathode during cycling against contact issues. Furthermore, as seen from the updated Fig. 5e whose y-axis have been enlarged to 60-110 %, the Coulombic efficiency fluctuations appeared in both Li|HGPE|LRO/graphite and Li|HGPE|LRO cells. Therefore, the fluctuation can be attributed to the structural evolution of the LRO during cycling rather than the contact issue caused by PF₆⁻ intercalation from/into the graphite. We have tested the longterm cycling performance of SRLMBs at 0.5 C. As shown in Supplementary Fig. 34, the capacity retention of Li|HGPE|LRO/graphite cell remained 67.8 % after 200 cycles, which is slightly higher than that of the Li|HGPE|LRO cells (63.4 %), indicating that this “shuttle-relay” chemistry can increase the battery specific energy without sacrificing its cycling stability. We have revised the Fig. 5e on Page 25 in the revised manuscript, and added Supplementary Fig. 34 on Page 18 in the revised Supplementary Information.

Fig. 5e Cyclic performances of Li|HGPE|LRO and Li|HGPE|LRO/graphite cells at 0.2 C.

Supplementary Fig. 34 Cyclic performances of Li|HGPE|LRO and Li|HGPE|LRO/graphite cells at 0.5 C.

3) As commented by the reviewer, we used two kinds of carbon as conductive agents in the cells, *i.e.*, 8 wt% KS6 graphite and 2 wt% CNT. KS6 graphite was applied to support the “shuttle-relay” chemistry, while CNT was introduced to optimize the electronic conductive network in the cells since the dispersibility of KS6 graphite is relative poorer than tradition conductive agents (*e.g.*, acetylene black) due to its higher tap density. As shown in Supplementary Fig. 47, we compared the performance of Li|HGPE|LRO cells with 2 wt% CNT only, 8 wt% KS6 graphite only and 2 wt% CNT with 8 wt% KS6 graphite as conductive agents, respectively. Only the cell using 2 wt% CNT with 8 wt% KS6 graphite exhibited a stable cycling. This clearly demonstrated that the synergistic effect of KS6 graphite and a small amount of CNT facilitated the construction of a fast and robust electronic conductive network. We have added above explanation on page 25 in the revised Supplementary Information.

Supplementary Fig. 47 Cyclic performances of Li|HGPE|LRO cells with 2 wt% CNT, 8 wt% KS6 graphite, and 2 wt% CNT with 8 wt% KS6 graphite at 0.2 C.

References:

- 1 Lin D, *et al.* *Nat. Nanotechnol.* **11**, 626-632 (2016).
- 2 Wang M, *et al.* *Advanced Materials* **29**, 1703882 (2017).

2) *Serious concerns are related to the HGPEs. Authors showed the polymer matrix as a Copolymer network. However, they have used a highly reactive acrylate and a very low reactive allylic phosphate and tried to copolymerize it together, expecting it to be a co-polymer network as demonstrated in the scheme. Due to the lower reactivity, allyl will polymerize slowly and separately. The acrylates due to their high reactivity, will polymerize very fast. This is also reflected in the results, for examples, they obtained a very low conversion rate of 71%. Acrylates as unreacted monomers will create issues with the lithium metal in a full cell. From polymerization point of view, Authors must sort out this polymerization issue and identify the polymer matrix structure in details. Authors should study the kinetics of acrylate and DAP polymerization separately. Indeed the polymerization time of 1 hour is not sufficient for DAP to form a polymer of significant molecular weight. In the best case scenario Authors may achieve sIPN/IPNs as the polymer matrix. Then the use of LiPF₆ and*

polymerization temperature are challenging combinations. Especially, the polymerization temp of 70 °C and PF6 stability are a serious issue. Indeed, at that temperature, acrylates might be polymerized by cationic polymerization induced by PF6 anions, which decompose into HF and PF5. Indeed, Authors didn't purify the monomers and used as it is, which will induce water molecules in the electrolyte and form various unwanted products. These issues are serious from the polymerization front. Also, it is not clear why the Authors used vacuum for polymerization and how are they going to control the evaporation of the solvents at 70 °C and vacuum. Such issues will create bubbles and uneven contacts during the in-situ processing. The AIBN decomposition is also different in vacuum, which will create further issues for the polymerization.

Response: Thanks for these valuable comments. A point-to-point response is described below:

1) The reviewer thought the polymer matrix in HGPE may be semi-interpenetrating polymer networks (sIPN)/ interpenetrating polymer networks (IPNs) due to the different polymerization kinetics of diethyl allyl phosphate (DAP) and pentaerythritol tetraacrylate (PETEA). We investigated the conversion rates of DAP and PETEA monomers by ¹H NMR spectra^[1]. The conversion rate of monomers can be estimated from the integrated area ratio of CH₂= on the monomers in the polymerized gel/solution to that in pristine precursor solution. The CH₂- on FEMC solvent was set as reference. As shown in Supplementary Fig. 6a-f, after a polymerization at 70 °C for 60 min, the conversion rates of 3 wt% DAP only and 1.5 wt% PETEA only in the liquid electrolyte (1 M LiPF₆-FEC: FEMC: HTE (1: 6: 3 by volume)) containing 0.1 wt% 2, 2'-azobis (2-methylpropionitrile) (AIBN) initiator were 15.6 % (Supplementary Fig. 6a-b) and 90.0 % (Supplementary Fig. 6c-d), respectively. Notably, after heating the precursor solution containing 3 wt% DAP and 1.5 wt% PETEA, the conversion rate of DAP increased to 32.4 % and that of PETEA decreased to 81.9 % in the HGPE, indicating a copolymerization of these two monomers (Supplementary Fig. 6e-f). Moreover, by varying the monomer ratio in the AIBN-containing electrolyte (the total monomer amount was set as 4.5 wt%), the competitive polymerization rate can be obtained following the formulas below^[4]:

$$\frac{d[M_{DAP}]}{d[M_{PETEA}]} = \frac{[M_{DAP}](r_{DAP}[M_{DAP}] + [M_{PETEA}])}{[M_{PETEA}]} \quad (1)$$

Where the r_{DAP} and r_{PETEA} is the competitive polymerization rate of two monomers; $[M_{DAP}]$ and $[M_{PETEA}]$ are the

concentrations of two monomers; $\frac{[M_{DAP}]}{[M_{DAP}] + [M_{PETEA}]}$ is composition of copolymer, which is estimated by ratio the integral peak area of two monomers. We define $P = \frac{[M_{DAP}]}{[M_{DAP}] + [M_{PETEA}]}$ and $R = \frac{[M_{DAP}]}{[M_{DAP}] + [M_{PETEA}]}$

$$R - \frac{R}{P} = \frac{R^2}{P} r_{DAP} / r_{PETEA} \quad (2)$$

$\frac{[M_{DAP}]}{[M_{DAP}] + [M_{PETEA}]}$, combing formula (1) to obtain formula r_{DAP} and r_{PETEA} can be obtained from the slope and intercept of a $(\frac{R}{P} - \frac{R^2}{P})$ plot, respectively. It is seen that r_{DAP} is 2.86 while r_{PETEA} is 0.06 (Supplementary Fig. 6g-h), suggesting that the DAP and PETEA monomers are

copolymerized and the matrix of HGPE is a block copolymer rather than sIPN/IPN^[2, 3]. This can be further confirmed by the polymerization phenomenon in Supplementary Fig. 6i, in which the gelation time of 1.5 wt% PETEA with 3 wt% DAP in AIBN-containing liquid electrolyte is shorter than that of the 1.5 wt% PETEA in AIBN-containing liquid electrolyte, verifying a copolymerization of these two monomers. We have added above result on page 31 in the revised Manuscript, and page 7 and page 57-58 in the revised Supplementary Information.

Supplementary Fig. 6 a, c, e ¹H NMR spectra and **b, d, f** corresponding conversion rates of **a, b** 1.5 wt% PETEA only, **c, d** 3 wt% DAP only and **e, f** 1.5 wt% PETEA with 3 wt% DAP in 1 M LiPF₆-FEC: FEMC: HTE liquid electrolyte containing 0.1 wt% AIBN at different polymerization time. **g** ¹H NMR spectra of 4.5 wt% monomer in liquid electrolyte containing 0.1 wt% with different DAP: PETEA ratios after polymerizing at 70 °C for 60 min. **h**

The corresponding $(R - \frac{R_1}{P}) - \frac{R_2}{P}$ plot. **i** The optical images of 1.5 wt% PETEA only (left), 3 wt% DAP only (middle) and 1.5 wt% PETEA with 3 wt% DAP (right) in liquid electrolyte containing 0.1 wt% AIBN at different polymerization time.

2) The reviewer thought that the polymerization time of 1 hour was not sufficient for DAP to form a polymer of significant molecular weight. Actually, the residual DAP monomer after polymerization acts as a cathode electrolyte interphase (CEI)-forming additive in the HGPE to further inhibit the electrolyte oxidation and the co-intercalation of solvent molecules into graphite. Furthermore, the reviewer thought unreacted PETEA monomer

will create issues with the lithium metal in a full cell. Indeed, due to the high conversion rate of PETEA monomer (81.9 %) in the HGPE, the effect of residual PETEA monomer on the battery performance was negligible, which is evidenced by the good cycling performance of both Li|HGPE|KS6 graphite and Li|HGPE|LRO/graphite cells.

3) The reviewer is concerned that a small amount of LiPF₆ salt decomposition may occurs at 70 °C to generate PF₅, which introduces a cationic polymerization of monomers and forms various unwanted products^[4]. Actually, the polymerization time at 70 °C was short due to the high reactivity of PETEA. Therefore, the decomposition of LiPF₆ was negligible. We have compared the gelation time of 1.5 wt% PETEA with 3 wt% DAP in FEC: FEMC: HTE electrolyte containing 1 wt% AIBN with 1 M LiPF₆, 1 M LiTFSI salt and without any salt. The difference in gelation time was negligible, demonstrating the decomposition of LiPF₆ did not obviously affect the polymerization process.

Figure R1. The gelation time of 1.5 wt% PETEA with 3 wt% DAP in FEC: FEMC: HTE electrolyte containing 1 wt% AIBN with 1 M LiPF₆, 1 M LiTFSI salt and without any salt.

4) We apologize that the description of the HGPE preparation was not clear. The preparation of precursor solution was in a glove box under atmospheric pressure, and then the precursor solution was sealed in the bottles/cells and heated by a vacuum oven at 70 °C. Therefore, the external vacuum environment did not affect the internal gelation process. We have added above explanation on page 31 in the revised Manuscript.

References:

- 1 Zhao Q, *et al. Nat. Energy* **4**, 365-373 (2019).
- 2 Creutz S, *et al. Macromolecules* **30**, 6-9 (1997).
- 3 Kim J-M, *et al. Scientific Reports* **4**, 4602 (2014).
- 4 Liu F-Q, *et al. Sci. Adv.* **4**, eaat5383 (2018).

3) *PF₆ interrelation leads to a change in PF₆ anion concentration in the electrolyte. How this issue is going to affect the ionic conductivity, and overall voltage of the cell during the initial and final stages of charge /discharge processes? Did the Authors study the influence of salt concentration in the electrolyte?*

Response: We have calculated the concentration change of PF₆ anions during the initial and final stages of charge /discharge processes of Li|HGPE|LRO/graphite cells. The salt concentrations in electrolyte were 1 M and 0.92 M before and after PF₆ intercalation into graphite. As shown in Supplementary Fig. 1a, the ionic conductivity slightly increased from 1.99 to 2.05 mS cm⁻¹ at 25 °C during this process. Furthermore, it is well-known that the salt concentration could dramatically affect the performance of dual-ion batteries based on anion shuttling^[1]. As seen in Supplementary Fig. 1, when the LiPF₆ concentration was set as 0.5 M in the HGPE (labeled as “0.5 M-HGPE”), the Li|LRO/graphite cell showed poor cycling performance (Supplementary Fig. 1c) with large polarization (Supplementary Fig. 1b) although the electrolyte ionic conductivity was as high as 2.61 mS cm⁻¹ at 25 °C (Supplementary Fig. 1a), mainly due to the insufficient salt concentration to support the anion intercalation. When the salt concentration reached 1.5 M in the HGPE, the ionic conductivity sharply reduced to 0.49 mS cm⁻¹ at 25 °C owing to formation of excess ion pairs (Supplementary Fig. 1a), which deteriorated the cycling performance (Supplementary Fig. 1c) and caused the increase of the battery polarization (Supplementary Fig. 1b). Therefore, the salt concentration was optimized as 1 M in the HGPE in this work, which exhibited the highest cycling stability (Supplementary Fig. 1c) with the lowest polarization (Supplementary Fig. 1b). We have added the above result on page 2 in the revised Supplementary Information.

Supplementary Fig. 1 a Ionic conductivities of HGPE with different salt concentrations at 25 °C. b Charge-discharge curves and c Cyclic performances of Li|LRO/graphite cells using HGPEs with different salt concentrations.

Reference:

1. Placke T, *et al. Joule* **2**, 2528-2550 (2018).

4) *From the manuscript one can understand that Authors have used a Li/SS cell for ECSW. They used a wide scan range. But there are already passivation and other decomposition happening at the electrodes during the*

reduction process. For example, SS is involved in a reaction around -0.43 V according to the Authors. Then, how can we consider this electrode as a working electrode (inert working electrode) while studying the anodic stability of a GPE? Indeed, the polymer is also not the same after the reduction process as mentioned by the authors, the monomers, PF_6 , and solvents decompose to form SEI layer in the reduction stage. Besides, in figure 8, supporting information, Authors have randomly selected the oxidation stability values. It is not clear what is the standard limiting value they have used to determine the electrochemical stability. Also in most cases, only passivation is observed not a decomposition, thus while assigning stability values, one must be extra careful. Besides, the membranes are already applied for reduction in the same cell. Thus, the discussion on ECSW is superficial. Thorough analysis is needed.

Response: Thanks for the reviewer's valuable comments. We re-tested the electrochemical window of the electrolyte by linear sweep voltammetry (LSV) at a scanning rate of 5 mV s^{-1} on a three-electrode system. Platinum foil was used as the working electrode, while lithium foil was used as the counter and the reference electrodes in this system. The oxidation potential values of electrolytes were recorded as the voltage when the current increased to $0.01 \text{ mA}^{[1]}$. As shown in Fig. 2f, low oxidation current was observed until 5.8 V for HGPE. The high electrochemical stability of HGPE mainly originates from the fluorination of the electrolyte solvent and the robust CEI formed by the oxidation of DAP at about 3.7 V (inset of Fig. 2f), which enables its application in 5 V -class

Li metal batteries. In sharp contrast, the irreversible oxidation voltages of $1 \text{ M LiPF}_6\text{-EC: EMC}$, $1 \text{ M LiPF}_6\text{-FEC: FEMC}$ and $1 \text{ M LiPF}_6\text{-FEC: FEMC: HTE}$ were around 3.8 V , 4.6 V and $5.2 \text{ V vs. Li/Li}^+$ (Fig. 2f and Supplementary Fig. 13), respectively. We have added above explanation on page 13, page 32 in the revised Manuscript, and page 8 in the revised Supplementary Information.

Figure 2f LSV curves of $1 \text{ M LiPF}_6\text{-EC: EMC}$ electrolyte and HGPE at a scan rate of 5 mV s^{-1} , using platinum foil as the working electrode and Li foil as the counter and reference electrodes.

Supplementary Fig. 13 LSV curves of the 1 M LiPF₆-FEC:FEMC and 1 M LiPF₆-FEC:FEMC:HTE electrolyte samples. The scan rate was 5 mV s⁻¹. Platinum foil was used as the working electrodes while Li foil was used as the counter and reference electrodes.

Reference:

1 Fan X, *et al. Nat. Nanotechnol.* **13**, 715-722 (2018).

5) *Authors must check for grammatical errors and spelling errors throughout the manuscript, for example: striping vs. stripping. Then in the abstract, first sentence is not very clear.*

Response: We have carefully checked and corrected the grammatical errors and spelling errors throughout the manuscript. The “striping” has been revised as “stripping” on throughout the revised manuscript, meanwhile the first sentence in the abstract has been revised as “Two promising measures have been predicated to increase the specific energy of current lithium (Li)-based batteries, including employing high-energy electrodes and utilizing capacities contributed by anion-shuttling.”

6) *Page: 4, line 73, construct a high-resistance cathode electrolyte interface (CEI) on the cathode surface, is it an interface or interphase?*

Response: We have uniformed it as “cathode electrolyte interphase”^[1] throughout the revised manuscript.

Reference:

1. Zhang Z, *et al. Matter* **4**, 302-312 (2021).

7) *Figure 2 f caption is wrong*

Response: We are sorry for our carelessness and have corrected the wrong figure caption.

8) Figure 2 b, FTIR curves must be provided before and after polymerization, otherwise readers cannot understand the difference.

Response: According to the reviewers' suggestion, we have added FTIR spectra of the precursor solution (before polymerization) and HGPE (after polymerization). As shown in Supplementary Fig. 7, the C=C at 1640 cm^{-1} almost disappear in the HGPE after polymerization, indicating a high conversion degree of monomers. Please see page 5 in the revised Supplementary Information.

Supplementary Fig. 7 FTIR spectra of precursor solution and HGPE.

9) Fig 23, CV peak behavior after the 5th cycle must be thoroughly discussed. currently, main peaks occurring after the cycling is mentioned, but not explained the reasons behind such changes.

Response: In the previous experiment, the cells were aged for 12 h before the CV testing and the initial CV curve was not repeatable with subsequent cycles (Supplementary Fig. 28a). We have extended the aging time to 24 h before testing, and found that the area of 1st CV curve increased while the 2-10 cycles were highly repeatable (Supplementary Fig. 28b). Therefore, we speculate the activation process in the 1st cycle is mainly due to insufficient wettability of HGPE to electrodes. The peak I in the CV curve represents both the formation of the corresponding graphite intercalation compound and the construction of the CEI film, meanwhile peak II and III indicates that PF₆⁻ anions successfully intercalate into the graphite layer, resulting in the generation of stage phases of graphite REF. Moreover, the de-intercalation peaks are in agreement with the intercalation peak, with a potential downshift of 0.1-0.3 V, indicating a reversible electrochemical behavior. We have added this result and discussion on page 15 in the revised Supplementary Information.

Supplementary Fig. 28 CV curves of the Li|HGPE|KS6 graphite cell after aging for **a** 12 h and **b** 24 h. The scan rate was 0.5 mV s^{-1} .

Reviewer #2:

In this work, the authors have demonstrated a high-performing “shuttle-relay” concept battery where capacity of graphite as a stand-alone or as an additive cathode material under high-voltages is reversibly utilized. The electrolyte utilized has been thoroughly characterized as well as its interaction with the anode and cathode during cycling. While the solutions employed here for a high-functioning “shuttle-relay” cell have individually been demonstrated ‘in-essence’, for example, the use of fluorinated solvents to get dense lithium metal deposition, use of fluorinated solvents with or without CEI-forming additives to get reversible capacity from graphite cathode, utilizing concentrated-electrolyte concept, etc, this work should be published in Nature Communications for its detailed characterizations with minor revision. Below are some comments for the authors:

- 1) Have the authors quantified the amount of residual DAP in their HGPE? It would be important to know as this residual DAP seems to be helping significantly in forming an effective SEI and CEI.

Response: Thanks for the reviewer’s positive comments on the quality of our work. The conversion rate of monomers was measured by ^1H NMR analysis with dimethyl sulfoxide- d_6 as solvent, which can be estimated from the integrated area ratio of $\text{CH}_2=$ on the monomers in the polymerized gel/solution to that in pristine precursor solution^[1]. The CH_2- on FEMC solvent was set as reference. As shown in Supplementary Fig. 6e and f, after polymerization, the conversion of DAP was 32.4 %, *i. e.* 2 wt% residual DAP in the HGPE facilitated the formation of stable SEI and CEI. We have added above result on page 9 in the revised Manuscript, and page 4 in the revised Supplementary Information.

Supplementary Fig. 6 e ¹H NMR spectra and f corresponding conversion rates of 1.5 wt% PETEA with 3 wt% DAP in 1 M LiPF₆-FEC: FEMC: HTE liquid electrolyte containing 0.1 wt% AIBN at different polymerization time.

Reference:

1 Zhao Q, *et al. Nat. Energy* **4**, 365-373 (2019).

2) Throughout the paper, the distinguishing cause behind some improvement of performance is almost always the residual DAP decomposition to improve SEI/CEI. What do authors think about putting DAP as an electrolyte additive and testing just the liquid electrolyte?

Response: According to the reviewers' suggestion, we have added 2 wt% DAP as electrolyte additive in the 1 M LiPF₆ in FEC: FEMC: HTE electrolyte. This concentration is equivalent to that of the residual DAP in the HGPE. As shown in Supplementary Fig. 4, the addition of DAP obviously improved the capacity retention of Li|1 M LiPF₆ in FEC: FEMC: HTE|LRO/graphite cells (from 81.9 % to 84.8 % after 50 cycles) due to the SEI/CEI-forming ability of DAP. Moreover, the cell with HGPE showed slightly higher cycling stability (96.9 % capacity retention after 50 cycles) than that of the cell using liquid electrolyte with DAP additive, because the polymer matrix inhibited the growth of lithium dendrites. We have added above result on page 3 in the revised Supplementary Information.

Supplementary Fig. 4 Cyclic performances of Li||LRO/graphite cells using 1 M LiPF₆ in FEC:FEMC:HTE electrolyte without/with 2 wt% DAP additive and HGPE at 0.2 C.

3) Page 32, line 570, the authors wrote: “The HGPE prepared via a facile in-situ.....and safety (i.e. non-flammable and free of liquid leakage).” I believe that the point “free of liquid leakage” has not been satisfactorily demonstrated especially during cycling. The data on pouch cell swelling in Fig 5g is not enough evidence of non-leakage as that might have just to do with low volatility of fluorinated solvents. Even the weight loss data in Supplementary Fig. 5 is not a direct evidence of leakage prevention during cycling. The authors can consider pursuing this further.

Response: Thanks for the reviewer’s valuable comments. We have performed leakage tests to compare the leakage of liquid electrolyte and HGPE^[1]. As shown in Supplementary Fig.10, we sealed 1 g 1 M LiPF₆ in FEC:FEMC:HTE liquid electrolyte and 1 g precursor solution of HGPE into Al plastic packages (55 mm× 50 mm), respectively, and heated the HGPE precursor to in-situ form HGPE. Then a small notch was cut in each package, and then the packages were squeezed for 15 s under a 1 kg weight. The liquid electrolyte package showed a 6 wt% leakage, while HGPE was absolutely leak-free. Subsequently, the packages were further hung for 5 min. The weight loss was 66 % for the liquid electrolyte, while the HGPE was as low as 0.7 %, demonstrating the superior resistance of HGPE against liquid leakage. We have added above result on page 6 in the revised Supplementary Information.

Supplementary Fig. 10 Leakage tests of 1 M LiPF₆ in FEC: FEMC: HTE liquid electrolyte (upper panels) and HGPE (lower panels).

Reference:

1 Ju J, *et al. Angew. Chem. Int. Ed.* **60**, 16487-16491 (2021).

4) Page 22, line 405, the authors wrote: “These are much higher than those of cells using 1 M LiPF₆-FEC: FEMC and 1 M LiPF₆-FEC: FEMC: HTE electrolytes (Supplementary Fig. 25).” I disagree that the performance of Li/HGPE/KS6 cell is significantly better than that of 1M LiPF₆-FEC: FEMC: HTE liquid electrolyte shown in SI Fig 25. The performance of these two are actually quite similar. And especially since the cathode loadings are very low (1 mg/cm²), some error in the weight measurements can cause the amount of difference in capacity seen between HGPE and the fluorinated liquid mixture shown in SI Fig 25, 26. Please rephrase this sentence if makes sense.

Response: Thanks for the reviewer’s valuable comment. We have rephrased the statement as “these (capacity values) are similar to those of the cells using 1 M LiPF₆-FEC: FEMC: HTE electrolytes, indicating that the gelation did not sacrifice the rate performance of batteries” on page 21 in the revised manuscript.

5) Page 18, line 340, there is a typing error. “(Supplementary Fig. 18, insets)” should be “Supplementary Fig. 19”.

Response: We apologize for this typo error and have corrected this on page 17 in the revised manuscript.

6) Page 18, line 337, please confirm whether the XPS analysis was performed on the cycled TEM grids or not.

Response: The Cu foils after plating-stripping 1 mAh cm⁻² lithium for 10 cycles at 0.2 mA cm⁻² rather than cycled TEM grids were subjected to in-depth XPS characterization. We have added this on Page 32 in the revised manuscript.

7) *In the caption of Supplementary Fig. 5, the readers are directed to Supplementary Note 3. The Note 3 is unrelated. Please check.*

Response: We apologize for this mistake. In the previous version, the Supplementary Note 3 is actually related to Supplementary Fig. 16-17 rather than Supplementary Fig. 5. We have revised the caption of Supplementary Fig. 16-17 to direct readers to Supplementary Note 4 (Supplementary Note 3 in the previous version). Please see page 9-10 and page 32-33 in the revised Supplementary Information.

REVIEWERS' COMMENTS

Reviewer #1 (Remarks to the Author):

Dear Authors,
the Authors have convincingly addressed the concerns raised by the reviewer. Thus, I would like to recommend the manuscript for publication in Nature communications in its present form. Good luck.
Best regards,

Reviewer #2 (Remarks to the Author):

The authors' response is satisfactory.

Dear Editor Dr. Rinaldo Raccichini,

Thank you very much for considering our manuscript entitled “A synergistic exploitation to produce high-voltage quasi-solid-state lithium metal batteries” that has been submitted to **Nature Communications**. It is an honor for us that the reviewers have given positive comments on the quality of our work and the editor have provided very detailed and professional revision suggestions. Herein, we respond to the reviewers' comments and present a point-by-point revision to the editor's suggestions. All revisions of format modification are highlighted by yellow background in the revised manuscript. We hope that this revised manuscript could meet the requirements for publication on your prestigious journal.

Thank you very much for your considerations. We are looking forward to hearing from you.

Best regards,

Guoxiu Wang on behalf of all co-corresponding authors and co-authors.

Response letter

Reviewer #1 (Remarks to the Author):

The Authors have convincingly addressed the concerns raised by the reviewer. Thus, I would like to recommend the manuscript for publication in Nature communications in its present form. Good luck.

Response: Thanks for the reviewer's positive comments on the quality of our work.

Reviewer #2 (Remarks to the Author):

The authors' response is satisfactory.

Response: Thanks for your positive comment on our revision.